# Feature-based reward learning shapes human social learning strategies

**David Schultner** [1] ✉, **Lucas Molleman** [2] **& Björn Lindström** [1] ✉

Human adaptation depends on individuals strategically choosing whom to learn from. A mosaic of social learning strategies—such as copying majorities or successful others—has been identified. Influential theories conceive of these strategies as fixed heuristics, independent of experience. However, such accounts cannot explain the flexibility and individual variability prevalent in social learning. Here we advance a domain-general reward learning framework that provides a unifying mechanistic account of pivotal social learning strategies. We first formalize how individuals learn to associate social features (for example, others' behaviour or success) with reward. Across six experiments ($n = 1,941$), we show that people flexibly adjust their social learning in response to experienced rewards. Agent-based simulations further demonstrate how this learning process gives rise to key social learning strategies across a range of environments. Our findings suggest that people learn how to learn from others, enabling adaptive knowledge to spread dynamically throughout societies.

Social learning—or learning from others—is key to the complexity of human culture and the success of humans as a species[1–4]. However, not all social information is equally useful. For social learning to be adaptive and in turn allow culture to accumulate, individuals need to use it strategically[5–8]. Since the 1980s, research has documented a mosaic of at least 26 social learning strategies, including strategies such as 'copy the majority' and 'copy successful others'[9–12]. Although this research highlights which forms of social learning may facilitate adaptive behaviour, the mechanisms underlying these strategies remain unclear. Here we demonstrate that a simple domain-general reward learning mechanism provides a unifying explanation for key parts of this disjointed set of social learning strategies.

Models of cultural evolution build on mathematical tools from evolutionary biology to analyse how individuals' social learning strategies shape collective dynamics[2,3,13–15], providing an influential perspective on how culture changes over time[16,17]. These models typically assume that social learning strategies are domain-specific heuristics, meaning they are specialized rules tailored to particular situations, which individuals employ in learning from others[18]. According to this view, individuals are equipped with a diverse array of such heuristics that are independent of experience[19]. Although the existence of social

learning strategies has been documented in a range of studies[6,8,11,20,21], several recurring empirical findings challenge the view that these strategies represent immutable fixed heuristics. First, experimental studies suggest that social learning is adaptable and can change with experience[22–26]. Second, the fixed heuristics view is unable to explain how conflicting strategies are resolved, for example, when different strategies suggest different behaviours[11,21]. Finally, this view does not account for the widely observed and substantial degree of between-individual heterogeneity in social learning[19,24,25,27]. For example, evidence from decision-making experiments shows that some people follow majorities, others focus on peer payoffs, and yet others ignore social information altogether[28,29]. In other words, the fixed heuristics account of social learning strategies suffers from both lack of parsimony and explanatory power.

An alternative perspective suggests that social learning instead rests on domain-general associative learning mechanisms[30–33]. According to this view (which we call the 'reward learning account' of social learning), social learning is not governed by fixed domain-specific heuristics. Instead, it is shaped by rewards and punishments so that individuals learn to use social learning based on their personal experience. For instance, the social learning strategy 'copy the majority'

[1]Department of Clinical Neuroscience, Karolinska Institutet, Stockholm, Sweden. [2]Department of Psychology, University of Amsterdam, Amsterdam, the Netherlands. ✉e-mail: david.schultner@ki.se; bjorn.lindstrom@ki.se

may be adopted if an observed majority indeed provides valuable guidance[30–33], and individuals may learn to 'copy the successful' if this has paid off in the past. Emerging evidence from neuroimaging[34–38], computational modelling[39–44] and machine learning[45–47] supports the idea that reward learning might be integral to social learning. Yet, while the reward learning account may offer parsimony, generalizability and flexibility, its power to explain human social learning strategies remains unclear. First, in contrast to the fixed heuristics account, the reward learning account so far rests on verbal rather than formal reasoning. Second, while the fixed heuristics account is supported by a host of experimental and field evidence, empirical support for the reward learning account remains limited[11].

Here we demonstrate formally and empirically that reward learning provides a mechanistic and parsimonious explanation for the emergence of social learning strategies. In brief, our findings demonstrate that people not only learn which actions result in rewards, but they also learn which features of the social environment are predictive of rewards. We show how, as a consequence, this process can give rise to behaviour consistent with specific social learning strategies (such as 'copy the majority' and 'copy the successful'). Our account reduces the diverse taxonomy of seemingly distinct social learning strategies into behavioural expressions of one single underlying learning mechanism. Furthermore, unlike heuristic accounts of social learning, our approach (1) allows individual variability in social learning strategies due to environmental influences, (2) allows for these strategies to change over an individual's lifetime in a way that generalizes across situations, and (3) resolves conflict between competing strategies.

We proceed as follows. First, we formalize the reward learning account with the social feature learning (SFL) model. This model, based on classic associative learning theory[48] and modern reinforcement learning[49], captures the idea that people learn about the value of social features, for example, others' actions or payoffs, as predictors of their own rewards. According to the SFL model, the same learning mechanism operates on both social and non-social features. Second, we empirically validate this learning model in six experiments with human participants ($n = 1,941$). These experiments demonstrate that social learning is shaped by experience and underpinned by the same mechanism as individual learning, as assumed by the SFL model. A combination of quantitative model comparison and out-of-sample generalization demonstrates that the SFL model provides a better account of social learning than the fixed heuristics account and alternative learning models[42]. Finally, we employ model analysis and agent-based simulations to demonstrate that the SFL model can explain the emergence of key social learning strategies in a range of environments. We show that the SFL model modulates social learning in ways consistent with previous theory and empirical findings when varying key aspects of the environment, such as spatial variability or the prevalence of danger. In essence, we demonstrate that many of the hitherto unconnected social learning strategies emerge naturally when individuals 'learn' to learn from others.

Together, these findings put forward a simple, mechanistic and unifying model that builds on first principles of learning theory. This model parsimoniously accounts for a broad range of human social learning strategies and contributes to resolving outstanding challenges in cultural evolution research, including the within- and between-individual variability in social learning.

## Results
### Model overview
Our model posits that individuals learn which environmental features, both social and non-social, are predictive of rewards. This assumption aligns with long-standing ideas in associative learning theory[50–53] and neuroscience[54,55] that individuals perceive objects and situations as sets of distinct features, and learn about the values of these features rather than the whole objects and situations. This feature-based approach allows individuals to generalize their learning to novel situations that share similar features, even if they have not encountered those specific situations before. Our model applies this approach to learning about both non-social features (such as how something tastes) and social features (such as others' choices and outcomes).

To illustrate the basic idea, imagine an individual choosing between two actions (hunting rabbit or deer, Fig. 1a). Each action is associated with social (for example, how many others in the social environment hunt rabbit and deer) and non-social (for example, the size of an observed rabbit) features that might play a role in making the choice. The individual then makes a choice and experiences the outcome (reward or no reward, Fig. 1b) and uses this outcome to update the values of all the relevant features. This joint learning about social and non-social features of the world drives learning and decision making (Fig. 1c). Formally, the expected value, $Q$, of the action 'hunt rabbit', can be expressed as

$$Q(a = \text{hunt rabbit}) = s_{\text{rabbit}} w_{\text{rabbit}} + s_{\text{others}} w_{\text{others}} \qquad (1)$$

where $s$ indicates the feature values (for example, $s_{\text{others}} = 0.8$ if 80% of others in the social environment hunt rabbit) and $w$ indicates the weights reflecting the relative importance of each feature. These weights are learned from reward experience (Fig. 1c; see Methods for details). Real-world feature representations might be complex and high dimensional[56], but for the sake of exposition, we here consider a simple case in which the estimated value of an action is a linear combination of two feature values and the feature weights (that is, just as in linear regression).

Having estimated the values of the candidate actions (for example, 'hunt rabbit' and 'hunt deer'), the individual makes their decision using a standard softmax function, and the weights of the relevant features are updated on the basis of the obtained reward, using a version of the classic Rescorla–Wagner model (see Methods). Through this process, individuals learn the reward-maximizing values of the weights, non-social and social alike. In other words, individuals learn whether social features, such as others' actions or outcomes (or any other social feature), are predictive of reward. Over time, both social and non-social features that are more strongly associated with past rewards will acquire larger weights and consequently impact actions more than features with weaker reward associations. In other words, social learning is shaped by individual reward experience, eliminating any intrinsic distinction between individual and social forms of learning.

For example, if individuals learn that others' tendency to hunt rabbit positively predicts their own rewards, this social feature will gain a positive weight. In contrast, if they instead learned that others' tendency to hunt rabbit tends to be negatively related to personal rewards, this social feature will get a negative weight. Finally, if others' hunting choices are not predictive of personal reward, the social feature weight will become zero. Here, the social feature weight, $w_{\text{others}}$, regulates the extent to which individuals take others' actions into account when making decisions in the future. Because this model, in contrast to earlier models, assumes that individuals learn about the distinct features rather than about the action per se, this allows generalizing to new situations. For example, if $w_{\text{others}}$ is positive, this will make the individual more likely to copy the majority in any situation with a majority, while if $w_{\text{others}}$ is negative, the individual will be more likely to copy the minority.

Finally, when several features predict the same outcome, the size of the updates and the resulting learned weights of individual features will be reduced. The reason is that the weights of all features are updated with the same mechanism and on the basis of the same outcome (see Methods for details). For example, if two social features predict the same outcome equally well, their respective learned weights will be reduced by half compared with scenarios where only one of the features

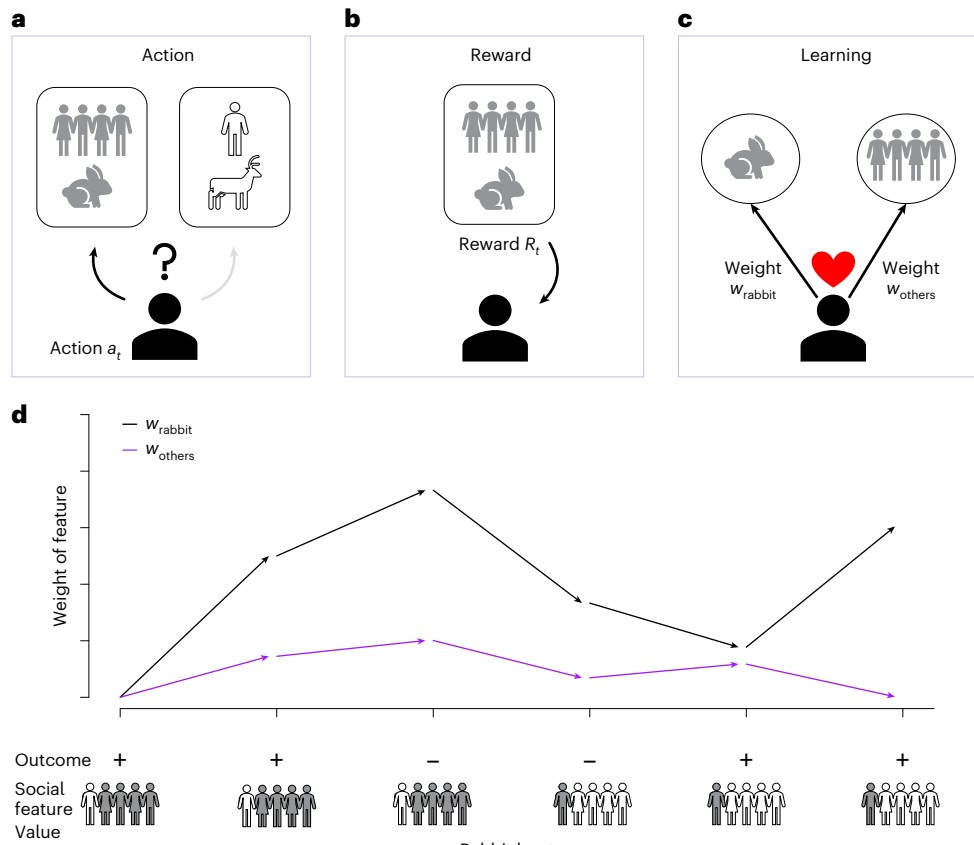

**Fig. 1 | The social feature learning model.** The SFL model assumes that people make choices based on the value of both social and non-social features associated with different actions (denoted with boxes), and update the weight of these features based on the experienced outcome. Social and non-social features are processed by the same learning mechanism. For example: **a**, when deciding what animal to hunt (rabbit or deer), individuals can consider both the size of the rabbit (a non-social feature) and how many others in their social environment hunt rabbit vs deer (a social feature). **b**, The individual receives a reward $R$ following their choice, and **c**, learns about both the non-social (size of the rabbit, $w_{rabbit}$) and social (others' choices, $w_{others}$) features of the action based on this outcome (indicated by the heart icon) by updating the respective feature weights. **d**, Example dynamics of the feature weights across six rabbit hunts. For the first three hunts, the majority in the individual's environment (indicated by the four grey stick figures) also hunted rabbit, while for the last three hunts, the minority (indicated by the single grey stick figure) hunted rabbit (and the majority hunted deer, white stick figures). The arrows indicate the direction of the weight update following the outcomes (+ indicates reward, − indicates no reward). Individuals update the weights of both features by the same outcome using the same mechanism of reinforcement. Note that the direction of the updates is the same for the non-social and social features when the majority hunts rabbit, but in opposite directions when the minority hunts rabbit.

is present. This process is known as 'feature competition' in associative learning theory and allows for approximation of the partial correlation between each feature and the outcome, removing the effect of other features[57].

In summary, the SFL model provides a direct formalization of the reward learning account of social learning strategies[30–33]. The model rests on two key assumptions: (1) social learning is shaped by individual reward experience, and (2) the same learning mechanism operates on non-social and social features of the environment.

### Empirical tests of the SFL model

Are these core assumptions of the SFL model realistic? To test this, we conducted a series of six incentivized experiments ($n = 1,941$) and evaluated these assumptions both qualitatively and with computational model fitting (see Methods for details).

The experiments were inspired by classic studies in cultural evolution research designed to test formal models of social learning under tightly controlled conditions[12,24,58]. Accordingly, the purpose of the experiments was not to mimic the real world, but to allow theory testing. Participants were incentivized to maximize reward by repeatedly choosing between different options represented by different colours (non-social features) and probabilistically receiving rewards.

These choices were made in the presence of social features (others' choices or others' payoffs).

**Social learning is shaped by individual reward experience.** In Experiments 1–5, we tested the first key assumption of the SFL model (see Fig. 2 for an overview and Supplementary Fig. 1 for a detailed task description). Experiment 1 provides an initial test of the hypothesis that social learning from others' choices is shaped by experience. Experiment 2 builds on this by examining a different social feature, focusing on learning from others' payoffs rather than their choices. In Experiment 3, we test whether among multiple social features, only reward-predictive features guide behaviour. Experiment 4 extends these findings to a more complex, four-option decision-making environment. Finally, Experiment 5 tests whether learning about social features generalizes to a novel, dissimilar task.

The basic experimental design involved two between-subject conditions: congruent and incongruent. Participants first completed a learning phase, in which they could learn whether social information was reward predictive (Fig. 2a). Depending on the condition, the social feature and reward were either aligned or misaligned: in the congruent condition, the high-reward option was chosen by the majority (or provided the most reward to others), while in the incongruent condition, it

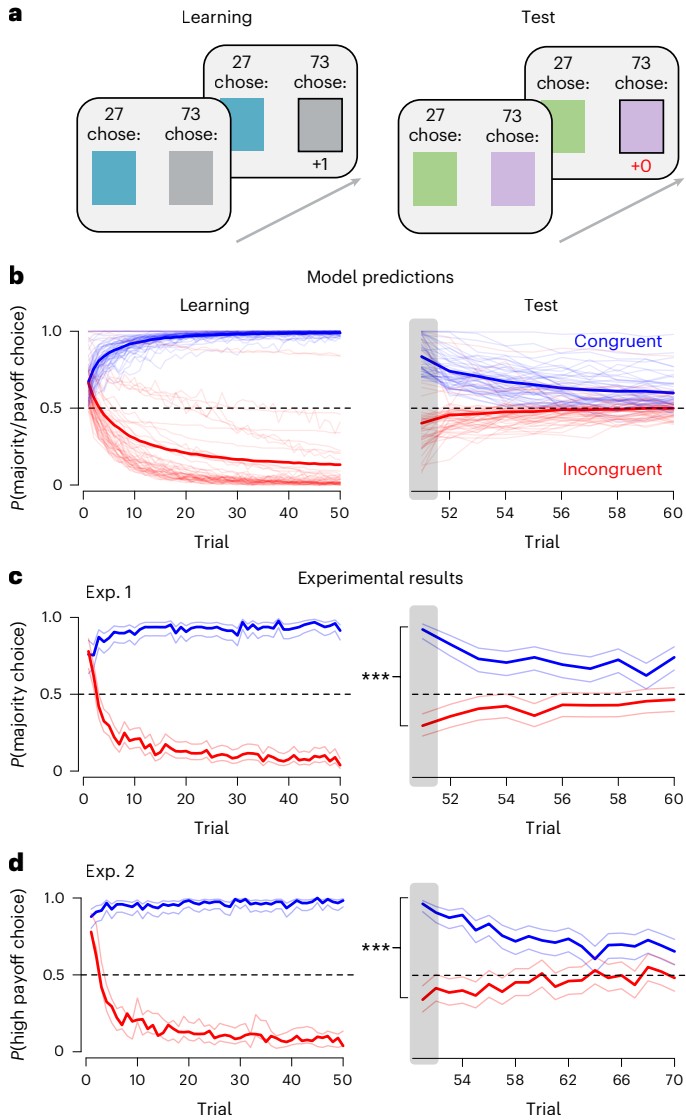

**Fig. 2 | Social learning is shaped by individual reward experience. a**, Participants completed a probabilistic decision-making task. Choice options had non-social (square colour) and social features (Exp. 1: others' choices (as illustrated in **a**); Exp. 2: others' payoffs). During the learning phase, participants could learn which of two options was most rewarding. Social features and rewards were aligned in the congruent condition and misaligned in the incongruent condition. The subsequent test phase was used to assess whether social learning was shaped by rewards; participants were exposed to two new options (different coloured squares). **b**, SFL model predictions (identical for the two social features, see Methods) of how reward experience in the learning phase shapes social learning in the test phase (faint lines, individual simulation runs; solid lines, averages; see Methods for details and Supplementary Fig. 2 for simulations across a range of parameters). The model includes a weight prior parameter, which allows for a majority preference on the first trial of the learning phase. **c,d**, Empirical results closely align with the SFL model, with lines showing averages with 95% CIs around the condition mean. **c**, In Experiment 1, participants learned to follow the majority (minority) in the congruent (incongruent) condition in the learning phase. **d**, In Experiment 2, participants learned to follow (avoid) the option with high payoffs for others in the congruent (incongruent) condition in the learning phase. For both experiments, social learning spilled over into the test phase. We tested the between-condition difference on the first test phase trial using logistic regression: Exp. 1 $p < 0.001$, $n = 285$; Exp. 2 $p < 0.001$, $n = 244$. Grey boxes highlight predictions at the outset of the test phase; ****p < 0.001$.

was chosen by the minority (or gave the least reward to others). According to the SFL model, experiencing congruent social features will lead to positive social feature weights, while experiencing incongruent social features will lead to negative social feature weights. In the subsequent test phase, participants faced new choice options with equal reward probability; these options were characterized by the same social feature the participants just experienced, but novel non-social features. Participants' initial choices in the test phase revealed whether social feature weights were shaped by rewards experienced in the learning phase. In Experiment 5, the test phase was replaced with an estimation task in which participants could use social information to improve their decisions.

In Experiment 1 ($n = 285$), we investigated whether social learning from others' choices (social feature) is shaped by reward experience (cf. majority bias or conformity[24,59,60]). As an initial step, we first evaluated whether participants showed reward learning in the learning phase. In both the congruent and incongruent condition, participants learned to select the optimal option (Fig. 2c; last 10 trials of learning phase, logistic regression, test against 0.5 in congruent condition: $\beta = 3.94$, s.e. = 0.27, $z = 14.71$, two-tailed $p < 0.001$, 95% CI [3.41, 4.46]; incongruent condition: $\beta = 3.23$, s.e. = 0.20, $z = 15.76$, two-tailed $p < 0.001$, 95% CI [2.93, 3.62]). Subsequently, participants in both conditions completed a 10-trial test phase where they chose between two novel options while observing others' choices. To test whether social learning was shaped by experience, we assessed the influence of others' choices on the first test phase trial. The SFL model predicts that individuals in the congruent condition will copy the majority, having learned that others' choices were reliable predictors of reward (Fig. 2b). Participants in the incongruent condition should instead copy the minority, having learned that others' choices were poor predictors of reward (Fig. 2b).

As predicted by the SFL model, social learning sharply differed by condition (Fig. 2c; grey boxes; between-condition difference (from congruent to incongruent) in copying the majority-aligned option on the first trial: logistic regression, $\beta = -3.33$, s.e. = 0.37, $z = -8.89$, two-tailed $p < 0.001$, 95% CI [−3.92, −2.61]). Participants in the congruent condition copied the majority (logistic regression, test against 0.5: $\beta = 2.45$, s.e. = 0.33, $z = 7.42$, two-tailed $p < 0.001$, 95% CI [1.86, 3.10]), whereas participants in the incongruent condition copied the minority (logistic regression, test against 0.5: $\beta = -0.87$, s.e. = 0.178, $z = -4.86$, two-tailed $p < 0.001$, 95% CI [−1.22, −0.52]; see Supplementary Information Section 2.3 for additional analyses). Overall, 92% of participants in the congruent condition followed the majority on the first trial, compared to only 30% in the incongruent condition. This indicates that social learning was shaped by experience, providing qualitative support for the SFL model.

The SFL model assumes that the same learning process can apply to any social feature. To test this assumption, Experiment 2 ($n = 244$) used a different social feature: we replaced others' choices from Experiment 1 with their average payoffs in a round (cf. payoff bias[61,62]). The results were again in agreement with the predictions of the SFL model (Fig. 2d; logistic regression, difference between conditions at the outset of the test phase: $\beta = -3.98$, s.e. = 0.55, $z = -7.30$, two-tailed $p < 0.001$, 95% CI [−5.06, −2.90]; congruent condition against 0.5: $\beta = 3.32$, s.e. = 0.51, $z = 6.53$, two-tailed $p < 0.001$, 95% CI [2.32, 4.32]; incongruent condition against 0.5: $\beta = -0.66$, s.e. = 0.20, $z = -3.36$, two-tailed $p < 0.001$, 95% CI [−1.05, −0.27]; see also Supplementary Information Section 2.3). These results demonstrate that both frequency-biased and payoff-biased social learning can be explained by the SFL model.

The SFL model posits that only reward-predictive features should guide social learning. In Experiment 3 ($n = 362$), we tested this prediction by making participants choose between options with two social features: others' choices and others' payoffs. Critically, only one feature correlated with reward. In the learning phase of the choice congruent condition, others' choices were reward predictive, while others' payoffs were not. Conversely, in the payoff congruent condition, others' payoffs were reward predictive, and others' choices were not (see Methods). After the learning phase, participants completed

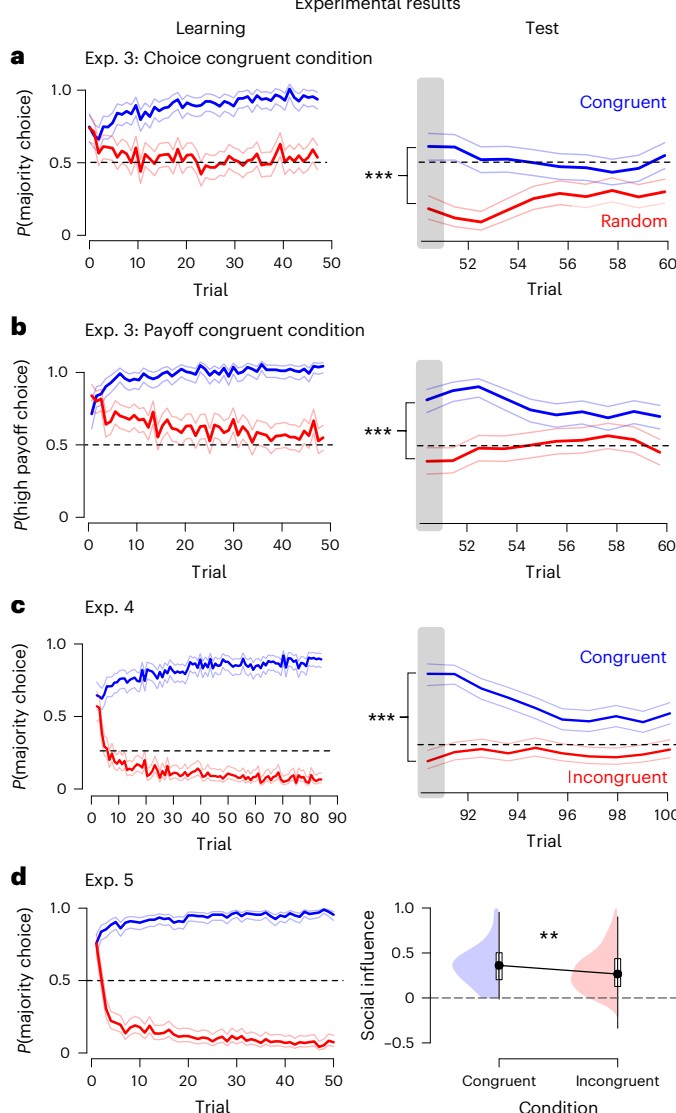

**Fig. 3 | Reward experience shapes social learning across settings. a**, In the choice congruent condition of Experiment 3, participants' social learning was guided by others' choices. **b**, In the payoff congruent condition of Experiment 3, social learning was shaped by others' payoffs. Logistic regression across both conditions shows a difference, $p < 0.001$, $n = 362$. **c**, In Experiment 4, participants selected between four options and learned to follow the majority (minority) in the congruent (incongruent) condition. Logistic regression was used to test for a difference, $p < 0.001$, $n = 353$. **d**, In Experiment 5 (social feature: others' choices), experience in the learning phase generalized to a social influence task (right), where participants in the congruent condition were more susceptible to social influence than those in the incongruent condition. The difference in social influence was tested using a Wilcoxon signed-rank test, $p = 0.004$, $n = 388$. The boxplots show the median point, their 1st and 3rd quartiles, and the most extreme values within $1.5 \times IQR$ (whiskers). All confidence bands represent 95% confidence intervals around the condition mean; **$p < 0.01$, ***$p < 0.001$.

a test phase where the two social features were pitted against each other: one option was chosen by the majority but yielded low payoffs to others, whereas the other option was chosen by the minority but yielded high payoffs to others. As predicted by the SFL model (Supplementary Fig. 3), participants preferred the option aligned with the reward-predictive social feature in the first trial of the test phase (Fig. 3a,b; logistic regression across the payoff congruent and choice congruent condition: $\beta = 1.54$, s.e. = 0.25, $z = 6.15$, two-tailed $p < 0.001$, 95% CI [1.05, 2.03]).

Experiments 1–3 demonstrate reward learning of social learning in a simple two-choice scenario. To evaluate the predictions of the SFL model in more complex choice situations[23,63,64], Experiment 4 featured two additional choice options. Participants ($n = 353$) chose between four different options during the learning phase. In the congruent condition, the four options were chosen by 65%, 15%, 15% and 5% of others, such that the majority choice was reward predictive (see Methods for details). This pattern was reversed in the incongruent condition, such that the options were chosen by 5%, 15%, 15% and 65% of others and the minority choice was reward predictive. The test phase introduced four novel choice options with 25% reward probability each, and which again were chosen by 65%, 15%, 15% and 5% of others.

We first generated predictions under the SFL model (Supplementary Fig. 3j). Mirroring the previous experiments, participants in the congruent condition should follow the majority whereas participants in the incongruent condition should follow the minority at the outset of the test phase. The results of Experiment 4 closely match these predictions: participants in the congruent condition chose the majority option more (73%) than participants in the incongruent condition (14%, logistic regression of difference between conditions: $\beta = 2.86$, s.e. = 0.33 $z = 8.6$, two-tailed $p < 0.001$, 95% CI [2.21, 3.51], Fig. 3c). Similarly, participants in the incongruent condition chose the minority option (50%) more than participants in the congruent condition (10%, logistic regression of difference between conditions: $\beta = 2.26$, s.e. = 0.35, $z = 6.53$, two-tailed $p < 0.001$, 95% CI [1.57, 2.95]). These results confirm that the validity of the SFL model extends to more realistic settings with a larger number of behavioural options[23,63,64].

In Experiment 5 ($n = 388$), we tested whether learning about social features would generalize to a new, dissimilar social situation. The purpose was to rule out that the flexibility of social learning hinged on the similarity of the test and learning phases in Experiments 1–4. As in Experiment 1, participants first completed a learning phase in which others' choices indicated either the high-reward or low-reward option (congruent/incongruent condition) and subsequently a different social influence task[65]. On each trial of this task, participants submitted an estimate of the number of items they saw on the screen. Afterwards, they were exposed to the estimate of another participant and could change their initial estimate. The degree of social influence was measured by the extent to which the initial estimate was modified after observing another person's estimate[65].

The SFL model predicts that participants in the congruent condition should be more influenced by others' estimates than those in the incongruent condition. Consistent with the SFL model prediction, participants in the congruent condition were 29% more susceptible to social influence (median = 0.36) than those in the incongruent condition (Fig. 3d; median = 0.27; $W = 21,150$, two-tailed $p = 0.004$, 95% [0.02, 0.12]; see Supplementary Information Section 2.4 for additional analyses). Moreover, participants' weight of the social feature ($w_{others}$), estimated by the SFL model at the end of the learning phase, predicted individual differences in this susceptibility (robust regression: $\beta = 0.2$, s.e. = 0.07, $t = 2.72$, two-tailed $p = 0.007$, 95% CI [0.06, 0.35]; see Supplementary Fig. 4). These results demonstrate that the learned weight of social features generalizes across distinct social situations, providing further support for the SFL model.

In conclusion, the results from Experiments 1–5 provide convergent support for the first core assumption of the SFL model: people learn the value of social features from experience, which in turn strongly impacts how they learn from others.

**The same learning mechanism operates on non-social and social features of the environment.** After Experiments 1–5 verified the first core assumption of the SFL model—people learn the value of social features from experience—the concluding Experiment 6 ($n = 309$) aimed to test the second core assumption of the SFL model: a single learning mechanism operates on both social and non-social features.

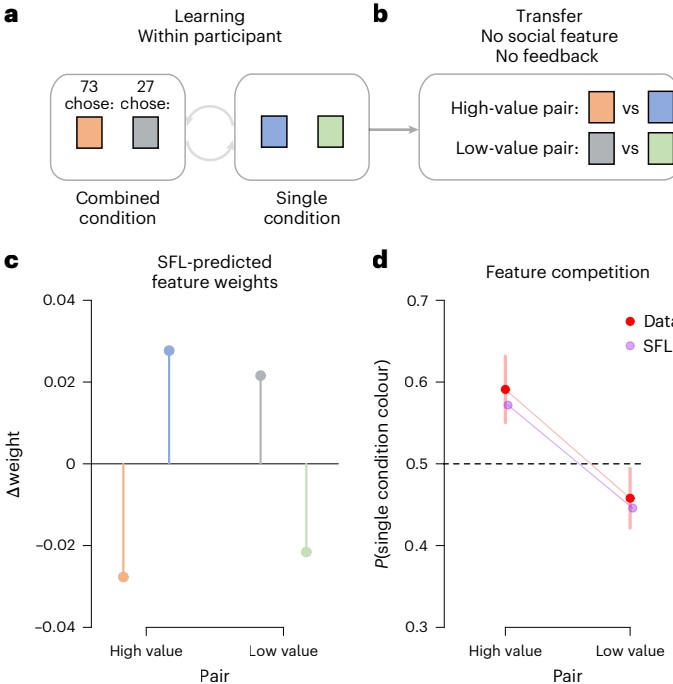

**Fig. 4 | The same learning mechanism operates on non-social and social features of the environment. a**, In Experiment 6, every participant completed two learning-phase conditions, combined and single, in counterbalanced order. Each condition involved choices between two options, indicated by different non-social features (colours). The combined condition also included reward-congruent social information (others' choices). **b**, In the transfer phase, the colours from the combined and single conditions were pitted against each other in a series of non-reinforced choices, in the absence of social features. **c**, SFL-expected Δ weights, after learning, for the non-social feature colours from the combined and single conditions, ordered in high-value and low-value pairs. To improve visibility of the pattern, the mean weight of each pair was subtracted from the individual weights. **d**, The transfer phase results verified the prediction of feature competition. Participants preferred the colour from the single condition in the high-value pair, and the colour from the combined condition feature in the low-value pair, matching the predicted pattern in **c**. Multilevel logistic regression compared to 0.5 for the high-value pair, $p < 0.001$, and for the low-value pair, $p = 0.035$, $n = 309$. Out-of-sample predictions from the SFL + prior model are based on median parameter estimates from Experiment 1. The error bar indicates 95% CI.

These learning mechanisms are not directly observable, so we devised a diagnostic task based on feature competition. Feature competition, a hallmark of associative learning theory[50,57,66,67], describes how learning about co-occurring features predicting the same outcome is reduced compared with learning about these features separately (see also refs. [33,68]). For example, if two distinct features (for example, a colour and others' actions) both predict the same outcome, their learning weights would be smaller than if each feature were learned about one at a time. Because the SFL model assumes that the same mechanism operates on both social and non-social features, it predicts that these features should compete with each other.

The task had two counterbalanced within-subject conditions and a transfer phase (Fig. 4a,b and Methods). The combined condition was equivalent to the congruent condition in Experiment 1: participants learned which of two colours (non-social features) yielded more rewards (80% or 20% of the cases) in the presence of others' reward-aligned choices (social feature). In the single condition, participants completed an equivalent learning phase with two different colours but without social information. The probability of selecting the high-reward option did not differ between the conditions (logistic regression: $\beta = 0.008$, s.e. $= 0.11$, $z = 0.075$, two-tailed $p = 0.94$, 95% CI

[−0.21, 0.22]), indicating no evidence for a difference in learning. In the subsequent transfer phase (Fig. 4b), colours from both conditions were pitted against each other, such that participants chose between the two colours in the pair of previously encountered high-reward colours as well as the two previously encountered low-reward colours. The colours were now presented without reward feedback to prevent further learning, and—crucially—without social features.

If learning about social and non-social features relies on the same mechanism as assumed by the SFL model, we should observe less learning about the non-social features (colours) in the combined condition than in the single condition. The reason is that in the combined condition, both social and non-social features predict rewards, so that feature competition impairs the learning about the colour weights. The expected consequence of this is a highly specific pattern of feature weights across the two conditions (Fig. 4c) after learning. The transfer phase (Fig. 4b) allowed us to directly measure these feature weights (see ref. [69] for a similar approach). Here, the SFL model predicts that participants, as a consequence of the feature weights (Fig. 4c), should assign more weight to, and therefore prefer, the colour from the single condition over the reward-matched colour from the combined condition in the high-value pair (Fig. 4d). They should also prefer the colour from the combined condition over the reward-matched colour from the single condition in the low-value pair (Fig. 4d).

The results confirmed these predictions: participants preferred the colour from the single condition over the equally reward-predictive feature from the combined condition in the high-value pair (multilevel logistic regression: $\beta = 0.618$, s.e. $= 0.14$, $z = 4.59$, two-tailed $p < 0.001$, 95% CI [0.34, 0.89]), and the colour from the combined condition ($\beta = -0.25$, s.e. $= 0.116$, $z = -2.11$, two-tailed $p = 0.035$, 95% CI [−0.48, −0.02]) over the colour from the single condition in the low-value pair (Fig. 4d; see also Supplementary Information Section 2.5). By establishing the presence of competition between social and non-social features, Experiment 6 supports the second core assumption of the SFL model, that the same learning mechanism operates on both social and non-social features.

Together, the results from the six empirical studies provide strong qualitative support for the SFL model as an account of human social learning. As assumed in the model, the results demonstrate that humans adjust social learning from experience, and use the same mechanism to learn about both individual and social features. We next turn to quantitative model comparison to directly contrast the SFL model against a range of alternative accounts of the mechanisms underlying human social learning.

## The SFL model outperforms alternative models

We used computational modelling to quantify how well the SFL model explains patterns in our data compared to two influential alternative accounts of social learning (Fig. 5): First, the fixed heuristics account which has been extensively analysed in previous experimental studies of social learning strategies[12,22,24,58]. This account, also known as decision biasing[23], typically assumes sensitivity to social information to be fixed within the individual, and lacks mechanisms for adapting social learning based on experience[12,22,24,58]. Second, the recently proposed value shaping account, which assumes that social observation functions as a pseudo-reward that directly influences expectations about the value of different options[42,44]. The degree of this influence depends on how well others' actions or outcomes align with the individual's own value estimates (see Methods for details).

Model comparison shows that the SFL model provides the most parsimonious account of the data across all six studies (protected exceedance probability = 1, Fig. 5a) and for each experiment separately (Supplementary Fig. 5). These results demonstrate that the choice of most participants were better described by the SFL model than the fixed heuristics and value shaping models. In addition to quantitative model comparison, we also tested the generalizability of the SFL model by

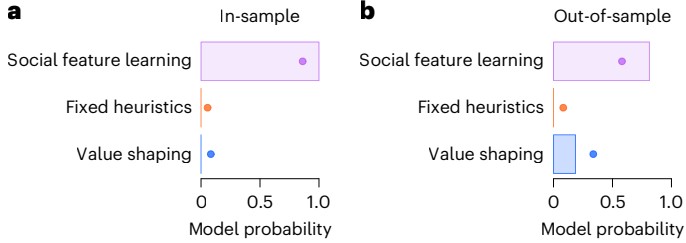

**Fig. 5 | In-sample model comparison and out-of-sample prediction. a**, The figure shows the estimated in-sample model probability (protected exceedance probability) of each model averaged across Experiments 1–6, representing the probability that this model is the best account of most participants' behaviour. Dots indicate the posterior model frequencies, the estimated prevalence of each model. See Supplementary Fig. 5 for model comparison separately for each experiment. **b**, Out-of-sample model probability and posterior frequencies averaged across Experiments 2–6, based on the median estimated parameter values from Experiment 1. See Methods for details.

generating out-of-sample predictions. This involved using parameter estimates derived from Experiment 1 (see Supplementary Table 1) to predict the data from Experiments 2–6 (see Methods). Employing such out-of-sample prediction provides a stringent test of a model's generalizability and robustness[70,71]. The SFL model provided the best overall out-of-sample predictions (protected exceedance probability = 0.81; Fig. 5b, see also Supplementary Fig. 6 for predictions for individual experiments and Supplementary Fig. 7 for out-of-sample predictions based on the mean, rather than median, parameter values), highlighting the SFL model's ability to generalize across different situations. Lastly, model simulations of all experiments revealed that only the SFL model could reproduce the key qualitative patterns in the data (Supplementary Fig. 3)[70].

Together, these results demonstrate that the SFL model provides a better account of social learning in our experiments than influential competing accounts. Having empirically verified the core assumptions and explanatory power of the SFL model, we next use model analysis and agent-based simulations to investigate whether the SFL model can provide a general account of social learning strategies.

### The unifying power of the SFL model

We first demonstrate how the core structural properties of the SFL model explain several key social learning strategies and how conflict between strategies can be resolved. Second, we implement the SFL model in agent-based simulations to assess whether it can explain the emergence of other key social learning strategies, and help explain both within- and between-individual variability in social learning. These two approaches correspond to analysing the individual (1) in isolation, and (2) in interaction with the environment.

### Model analysis

Both previous theory and empirical studies have established that people 'copy others when uncertain'[6,21]. The structure of the SFL model provides a simple mechanistic explanation for this social learning strategy. Intuitively, the relative influence of social features is amplified as individual uncertainty increases, where certainty is proportional to the absolute difference between the weights of the non-social features (Fig. 6a)[72,73]. For instance, in scenarios where individuals are highly certain about their preferences based on non-social features (for example, $w_{\mathrm{rabbit}} \approx 1$ and $w_{\mathrm{deer}} \approx -1$), social influence has little impact on their choices (Fig. 6a). However, in unfamiliar situations with higher uncertainty (so that $w_{\mathrm{A}} \approx w_{\mathrm{B}}$), the same individuals will be more influenced by social features (Fig. 6b), as social influence is maximized when uncertainty is highest (see Supplementary Fig. 8). In other words, when non-social features provide weak guidance,

social features gain relative importance for decision making as in the test phase in experiments 1–5 (Figs. 2 and 3). Conversely, when individuals have strong preferences, they are unlikely to be swayed by what others do, or any other social feature. Here, the SFL model reveals a tight mechanistic link between uncertainty-biased social learning and other social learning strategies.

By considering choice options as sets of weighted features, the SFL model also resolves conflicts between different social learning strategies, for example, when different strategies such as 'copy the majority' vs 'copy successful others' suggest different behaviours (Fig. 6c). Rather than assigning a weight to each distinct strategy (as has been proposed in the context of non-social decision making[74]), the SFL model assumes that social features with stronger past reward associations will receive larger weights and will therefore influence choice more than those with weaker associations. The results of Experiment 3 provide direct support for this structural aspect of the SFL, showing that people follow those features that tend to predict rewards (cf. Fig. 3a,b). In this way, apparent conflicts between different social learning strategies are resolved without requiring an extraneous arbitration mechanism[11].

Taken together, our model analysis shows how the structure of the SFL model provides a mechanistic explanation for key aspects of social learning. In a final step, we use agent-based models (ABM) to demonstrate how different social learning strategies emerge from the interaction of the SFL model with different environments.

### Agent-based simulations

Finally, we turn to agent-based model simulations to investigate the SFL model's potential as a unified, mechanistic explanation of social learning strategies. By simulating the SFL model, we test whether the same learning mechanism can lead to the emergence of distinct social learning strategies in different environments. These environments vary in aspects that previous models have shown to impact social learning strategies: they are temporally[75] and spatially variable[75], dangerous[6,20] or competitive[76]. The simulations mirror our experiments with a population of individuals aiming to maximize rewards in a multi-armed bandit setting. Simulated individuals make probabilistic choices on the basis of non-social and social features, experience outcomes, and update feature weights using the SFL model (Fig. 1).

We first consider the same features as in our experiments (others' actions and others' payoffs). We then extrapolate to other important social features (others' age and overall success)[77–80]. These features map to fundamental social learning strategies[11]. For the sake of exposition, we consider social features individually. Together, this yields a 4 (environment) × 4 (social features) simulation design (Fig. 7). Importantly, in these simulations, there are no inherited traits, and individuals start with no previous knowledge about the environment or social features (see Methods for details). We test what social learning strategies simulated individuals learn to use in the different decision-making environments. These strategies are indexed by the social feature weights, with higher feature weights indicating a stronger reliance on a specific type of social information. We also evaluate within- and between-individual variation in social feature weights.

In brief, these simulations show that the SFL model can explain the emergence of various key social learning strategies in a range of environments: when individuals adjust their reliance on social features on the basis of experienced outcomes, their emerging learning behaviour matches established social learning strategies, including copying the majority, payoff and prestige bias, and copying older individuals (Fig. 7). Furthermore, they demonstrate that both within- and between-individual differences in social learning can emerge directly from the stochastic nature of the learning process, offering a possible solution to the outstanding empirical puzzle of why this variation exists[19,24,28] (Fig. 8).

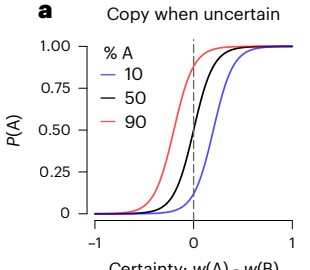

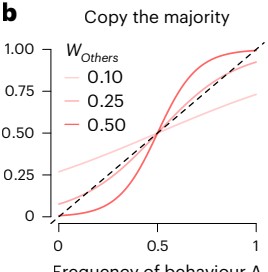

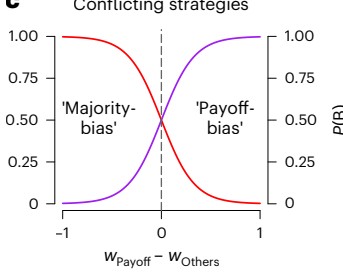

**Fig. 6 | The basic structure of the SFL model can account for multiple social learning strategies. a**, Copy when uncertain. The probability of selecting action A (for example, hunt rabbit) as a function of certainty (value difference of non-social features A and B) for three proportions of others choosing A. Social influence is strongest when A and B have similar weights (and uncertainty is largest; near the vertical dashed line). **b**, Copy the majority. The probability of selecting action A as a function of the frequency of others selecting A and $w_{Others}$, the learned weight associated with others' actions. **c**, Conflicting strategies. Conflict between different social learning strategies is resolved by the basic structure of the SFL model. The panel shows the probability of actions A (red) and B (purple), where A is associated with the majority choice (frequency A = 0.8) and low payoff of others (mean payoff = 0.2) and B with the minority choice (frequency B = 0.2) and high payoff of others (mean payoff = 0.8), as a function of the difference in the weight of the two social features ($w_{Others}$ and $w_{Payoff}$). Note how behaviour can be consistent with multiple distinct social learning strategies, depending on the relative weight of different features (labelled with 'Payoff bias' and 'Majority bias'). See text for details. In all panels, softmax $\beta$ = 0.1.

**Temporal variability.** A key consideration in evolutionary models of social learning strategies is that natural environments change over time[75]. These models demonstrate that the strength of majority bias (cf. conformity[24,59,60]) tends to decrease in more variable environments because others' actions become outdated sources of information. The SFL model recovers this insight (Fig. 7a): as the environment becomes more variable, individuals learn to put less weight on others' actions ($w_{Others}$). Notably, even a preference for the minority ('anti-conformity'[24]) is common in highly variable environments.

In the SFL model, similar trends are observed for the learned weight of both age ($w_{Age}$) and success features ($w_{Success}$) in variable environments (Fig. 7c,d). This occurs because the knowledge of older individuals becomes outdated when the environment changes (as demonstrated in previous evolutionary models[78]), and others' previous success becomes less predictive of future success. Conversely, in highly variable environments, individuals learn to assign more weight to others' payoff ($w_{Payoff}$), which aligns with previous theory indicating that payoff-biased social learning is most effective in changing environments (Fig. 7b)[61]. These first results demonstrate how four distinct forms of social learning strategies emerge from our simple, domain-general learning model.

**Spatial variability.** Individuals do not only experience temporal but also spatial variability when migrating between different natural environments. In the SFL model, increased spatial variability (defined as the probability to move between groups) leads to a higher weight assigned to others' actions ($w_{Others}$) and payoffs ($w_{Payoff}$) in our simulations (Fig. 7e,f). This closely aligns with previous evolutionary theories, which suggest that spatial variability promotes majority-biased social learning[60]. However, the pattern differed for age-biased social learning, as the weight assigned to others' age decreases with increased migration (when old age ceases to be an index of knowledge[78]), and for success-biased social learning, where the feature weight remained consistent (Fig. 7g,h).

**Dangerous environments.** According to previous theory, individuals should rely more on social learning when individual decisions risk being costly[11,20]. We investigate the impact of extreme costs, 'death', on the learned weight of social features[81]. To manipulate the risk of costly outcomes, we extended the simulation to include 8, rather than 2 options (while otherwise equivalent). Selecting some of these options resulted in the individual's death. To keep the population size constant, dead agents were replaced with new, naïve agents. As expected, we found that in the SFL, the learned weight of others'

choices ($w_{Others}$) was higher in more dangerous environments (Fig. 7i). In other words, when mistakes carry high costs, individuals learn to copy others more. In contrast, neither payoff-biased nor age-biased social learning was clearly related to the costs of individual learning (Fig. 7j,k), while dangerous environments reduced the reliance on others' success (Fig. 7l).

**Competition.** Resource competition is known to hinder the evolution of social learning[76]. In the SFL model, we observed that competition, which diminished the expected reward in proportion to the number of individuals adopting the same action (negative-frequency dependence), substantially decreased the average weight of all types of social features (Fig. 7m–p). This reduction occurred because, as competition intensified, social features became worse predictors of reward. When individuals distribute themselves due to competition[82], the social features of alternatives will tend to converge. For example, if an equal proportion of individuals selects each option, these options will have identical value of the feature related to others' actions. These findings underscore that the SFL model can offer a unified, mechanistic learning-based alternative to previous evolutionary theories of social learning strategies.

**Within- and between-individual variability in social learning.** The SFL model not only offers a unified, learning-based alternative to previous evolutionary theories but also sheds light on a central issue in cultural evolution research: the variability between and within individuals in their reliance on social learning[19,22,27,28,83]. According to the SFL model, both the within- and between-individual variability arise from the learning process.

To illustrate this point, consider a population in an environment with fluctuating rates of change (Fig. 8). As the environment becomes less predictable, individuals learn to rely less on social features, here exemplified with other's choices (see Supplementary Fig. 10 for an illustration with others' payoff as the social feature). This shift significantly impacts social learning strategies: in stable environments, individuals predominantly copy the majority (Fig. 8b), whereas in unstable environments, the same individuals are more inclined to copy the minority (Fig. 8c).

Crucially, there is also considerable variability between individuals (Fig. 8b,c), even when individuals learn in the same way and experience the same environment. This variability arises from random idiosyncratic experiences. For instance, some individuals learn that others' actions predict rewards well, while others learn to disregard them. In other words, an individual's learning history is crucial for their reliance

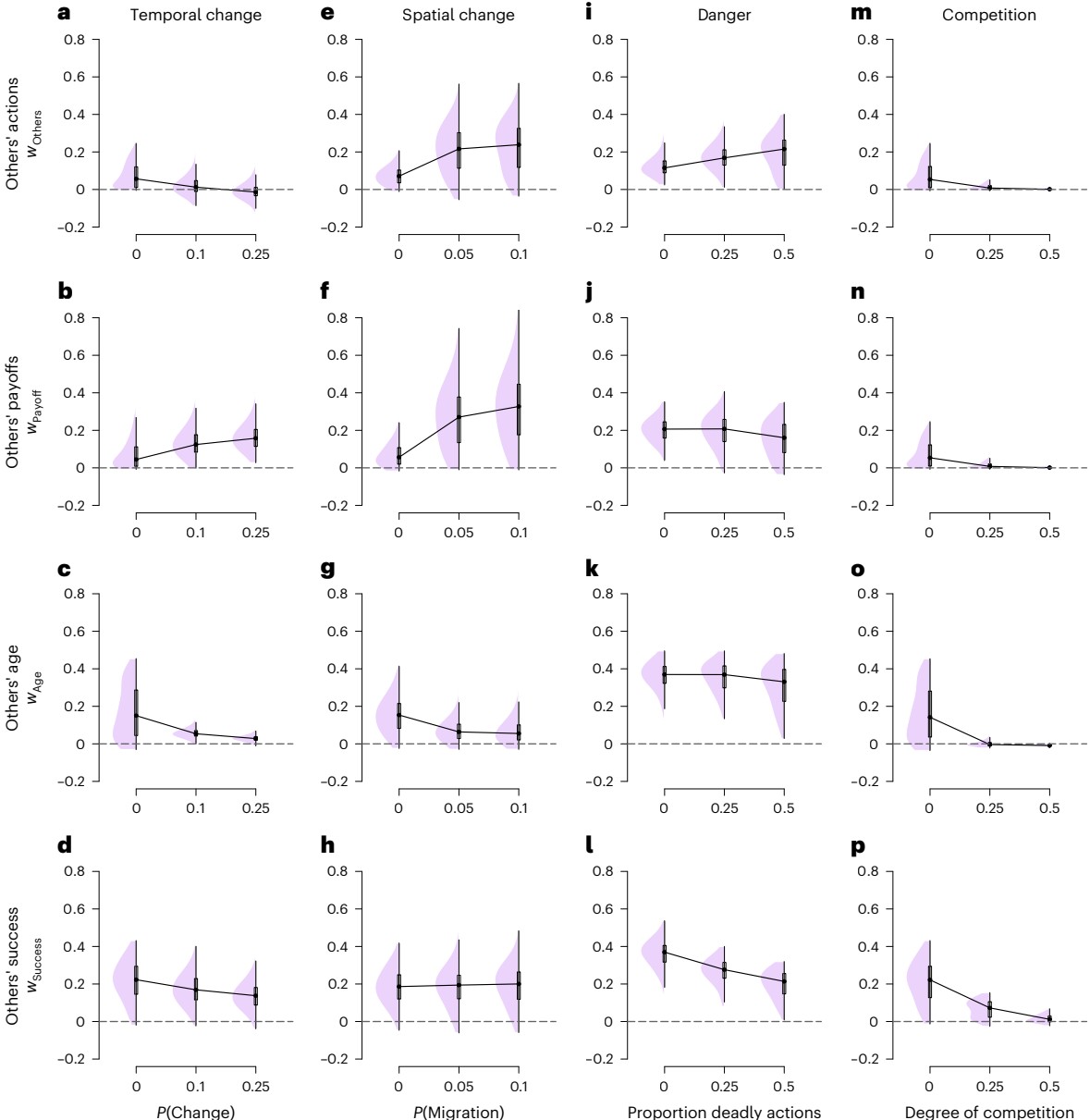

**Fig. 7 | Emergence of social learning strategies in agent-based simulations.** Learned social feature weights under temporal change (**a**–**d**), under spatial change (**e**–**h**), in dangerous environments (**i**–**l**) and with competition (**m**–**p**). Higher values correspond to a stronger influence. For all simulations, $N = 100$.

The figures show the median (black dot) average value from 800 simulation runs, the 'box' covers the first to third quartiles of the distribution, and the whiskers the range of simulation data.

on social learning. Notably, the between-individual variability is also larger in variable environments, since the relationship between others' actions and own rewards is then more stochastic (Fig. 8c).

In summary, the SFL model offers a possible solution to a long-standing puzzle in the study of social learning[19] – the notable variability observed both within and between individuals. According to the SFL model, this variability is attributed to two main factors: the characteristics of the learning environment, and random unpredictable outcomes, which result in different learning histories (see Supplementary Fig. 11 for a depiction of the empirical variability in our experiments).

## Discussion

We investigated whether social learning strategies, which are central for human adaptation, can be explained by domain-general reward learning mechanisms. We demonstrated that a reward learning account provides a parsimonious and mechanistic explanation of human social

learning strategies. According to this account, which formalizes previous verbal theorizing[30–33], social learning strategies emerge from individual reward experience. It connects a hitherto disconnected mosaic of social learning strategies by a common underlying mechanism. By uncovering how individuals learn to use social information, the SFL model helps address outstanding challenges for understanding human cultural evolution, including the within- and between-individual variability in social learning, and how conflicts between different social learning strategies are resolved.

Uncovering the mechanisms underlying social transmission has been described as one of the 'grand challenges' in the study of cultural evolution[84]. Our computational account of social learning strategies answers the calls for 'opening the black box' of social learning[31]. The SFL model offers an explicit algorithmic representation of how individuals process information, learning the predictive value of social 'features' such as others' actions or outcomes. This model – which is closely

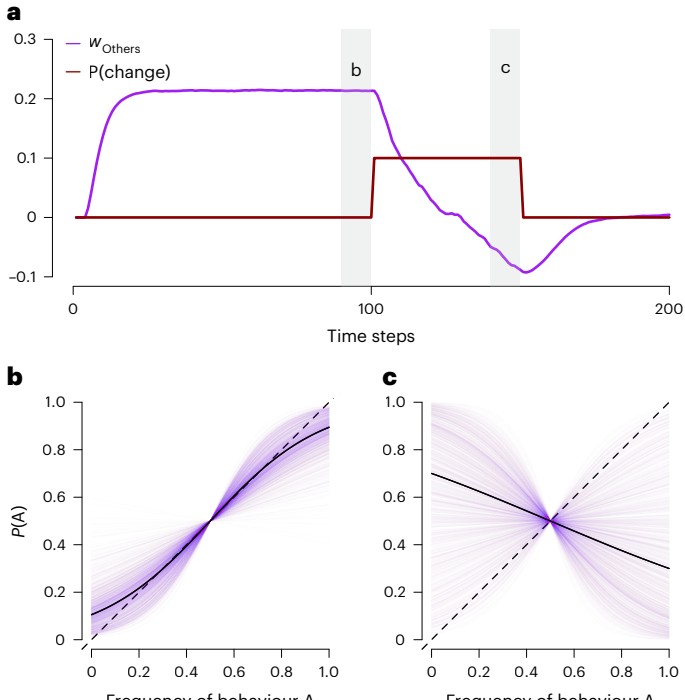

**Fig. 8 | Within- and between-individual variability in social learning is predicted by the SFL model. a**, On average, the social feature weight of others' actions ($w_{Others}$) decreases when the environment becomes less predictive of own rewards (while the payoff uncertainty is kept constant). The grey vertical bars indicate the time points depicted in **b** and **c**. **b,c**, Social influence functions. When making decisions between unknown options (where uncertainty is maximized, see Fig. 5a) the same (500, randomly chosen) individuals either copy the majority **b**, or minority **c**, depending on the current state of the environment in **a**. The figure also shows the considerable degree of between-individual variability expected by the SFL model. The solid black lines illustrate the average behaviour, while the dashed lines indicate random copying. Although we here exemplify behaviour using others' actions as the social feature, the same principle applies to any social feature (see Supplementary Fig. 10 for an example with others' payoffs).

aligned with standard models of value-based decision making[85], and both contemporary and classic learning theory[49,50]—presents reward learning as a foundational element for constructing a comprehensive and formalized theory of human social learning[39,86].

In a sense, the key difference between the SFL model and the standard fixed heuristics account is the time scale of adaptation. In the SFL model, learning from experience, rather than natural selection, is the driving force of optimization. Indeed, as our simulations demonstrate, the SFL model makes similar predictions about the impact of different environmental factors on social learning as the classic fixed heuristics view (Fig. 7). While these models have typically picked a few fixed heuristics and examined their evolutionary success[61,75], the SFL model adopts a single, explicitly mechanistic, reward learning-based approach. As learning operates on a much faster time scale than natural selection, this means that human social learning strategies are more flexible than commonly assumed.

In our experiments, participants quickly adapted their social learning strategies to changes in the reward associations of social features. These results suggest that the significant within- and between-individual variability commonly observed in social learning[19] may be driven by the domain-general learning mechanism captured by the SFL model. Based on this mechanism, individuals may develop and adopt a wide range of social learning strategies on the basis of their unique experiences and the specific rewards they encounter.

However, it should be emphasized that social feature learning does not exclude in-born biases in initial weights of social features, and how the features are processed. These biases might facilitate effective learning and are likely to be subject to natural selection[87–89]. The SFL model provides an account of how reward experiences may shape and refine these biases throughout development, attuning individuals' social learning to their environment.

The flexible nature of social learning strategies demonstrated in this study probably impacts the direction and outcome of cultural evolution. Outside of the lab, this could lead to greater diversity in cultural behaviours and practices within and across populations than previously expected on the basis of evolutionary models. Such issues could be addressed by incorporating social feature learning in dynamic models of cultural evolution. These models would provide insights into how factors such as the availability of resources, social norms and environmental contingencies shape individuals' learning experiences, and in turn determine which social learning strategies and cultural practices are reinforced. Another promising avenue of theoretical research would involve examining the consequences of social feature learning for cumulative cultural evolution. Diverse strategies may produce more cultural variation available for recombination to produce innovations, and when efficiency of a learning strategy can increase with experience, some individuals may become particularly proficient at innovating, and others may rapidly adopt their solutions (for example, because they learned what features are characteristic of good innovators).

There are several limitations to highlight. First, the SFL model focuses on how individuals learn to selectively use observable (social) cues to explain the emergence of strategic social learning. This means that the model does not directly address inferential forms of social learning, which are important for teaching[90,91]. Augmenting the SFL model with approaches such as inverse reinforcement learning[92], which can be used to infer the preferences or beliefs of others[93], could help address this limitation. Furthermore, in our implementation of the SFL model, we used simple feature representations. However, real-world feature representations are typically multidimensional[56], attention dependent[94] and often abstract[95], and updating of feature weights can be biased or moderated by the values of (other) features[96–98]. While it is possible to enhance the SFL model by incorporating more sophisticated feature representations (for example, with deep neural networks[99]) and asymmetric weight updating, the simplicity of linear features and unbiased updating enabled us to highlight the fundamental mechanistic principles of the model without sacrificing generality. It is also worth noting that the simple linear approximation sufficed for capturing human behaviour in our experiments (cf. Fig. 2), indicating that more complex representations may not always be advantageous.

In addition, our study employed simplified computerized experiments to study human social learning. Although this allowed us to design highly controlled settings where our theoretical model made distinct predictions[11], future research should further test the SFL model using non-experimental naturalistic data. Furthermore, our agent-based simulation setting, while comprehensive, did not cover the entire mosaic of social learning strategies. Future research could extend the SFL model to investigate how other important strategies, including preferences for familiar, prestigious or dominant others, can emerge through learning[79]. One interesting direction would be to explore the implications of individuals being able to explicitly represent feature weights and communicate them to others, allowing for the 'social learning of social learning'[19,31]. Moreover, although the experiments conducted here test scenarios with immediate feedback, the SFL model can naturally be extended to scenarios with delayed rewards: employing a temporal-difference updating rule[49] would allow social feature learning even when rewards are delayed by sequences of actions or by the passing of time. Finally, while our theoretical agent-based model simulations demonstrate that reward learning is sufficient for

social learning strategies to emerge in a population of naive individuals, confirmatory empirical evidence would require intensive developmental studies of the relationship between social information and rewards in children's everyday environments. In line with the reward learning account, existing studies suggest that reward learning plays an important role in the development of imitation skills among children[100]. However, more developmental work of human social learning would be needed to elucidate how social learning strategies are shaped by everyday experience.

In conclusion, our findings provide convergent theoretical and empirical evidence for domain-general learning as the key mechanism underlying human social learning strategies. Social learning strategies emerge naturally as a consequence when individuals 'learn' to learn from each other. Understanding social learning as an expression of reward learning provides a framework for studying the psychological and computational mechanisms underlying cultural evolution.

## Methods

### SFL model
We used $Q$-learning with linear function approximation[49] as the basis for the SFL model. This can be viewed as an extension of classic associative learning theory to instrumental behaviour and more complex (non-binary) representations of the environment. As described, the model learns the feature weights that maximize reward through repeated interaction with the environment. Features can be binary or continuous. The estimated value $Q(s, a)$ of action $a$ in state $s$ is a linear function of the features and their weights (equation 1). Decisions are made by comparing the estimated value $Q(s, a)$ of the actions (for example, 'hunt rabbit' and 'hunt deer') using a standard softmax policy function:

$$P(s, a_i) = \frac{e^{Q(s, a_i)/\beta}}{\sum_{j=1}^{k} e^{Q(s, a_j)/\beta}} \qquad (2)$$

where $j$ indexes all $k$ available actions, and $\beta$ [$\beta > 0$] determines the exploitation vs exploration balance. The SFL model updates the feature weights (that is, learns) through a standard Rescorla–Wagner update[49]:

$$\boldsymbol{w}_{t+1} = \boldsymbol{w}_t + \alpha(R_t - Q(s, a)) \, \boldsymbol{x}(a_t) \qquad (3)$$

where $R$ represents the reward, parameter $\alpha$ [$0 \le \alpha \le 1$] is the learning rate and $\boldsymbol{x}(a_t)$ the features associated with the chosen action. Here we use bold case to denote vectors. The basic SFL model has two free parameters, the learning rate $\alpha$ and the softmax temperature $\beta$. For the empirical analysis, we also considered a version of the SFL model (SFL + prior), where the initial value of the social feature weight was estimated as a free parameter, $P$. This version generally provided the best account of the empirical data. We also considered a version of the SFL model with separate $\alpha$ parameters for the non-social and the social feature, but this did not improve model fit.

### A priori model predictions
To generate a priori model predictions for Experiment 1 (Fig. 2b,c), we simulated the SFL model 10,000 times. We randomly drew parameter values for each simulation run from a uniform distribution, where the learning rate $\alpha = U(0.001, 0.4)$, softmax temperature $\beta = U(0.0001, 0.2)$ and prior weight $P = U(0, 0.2)$. See Supplementary Fig. 2 for simulations based on a larger range of parameter values.

### Experimental participants
Participants in all experiments (total $N = 1,941$) were recruited through Prolific. Participants were recruited globally, with fluent English language comprehension as the only recruitment criterion. Among participants who submitted their age, the mean age was 33.8 years (s.d. = 11.96), distributed across experiments as follows: Experiment 1

($M = 28$, s.d. = 9.4), Experiment 2 ($M = 33.6$, s.d. = 10.97), Experiment 3 ($M = 38.52$, s.d. = 13.89), Experiment 4 ($M = 36.39$, s.d. = 12.20), Experiment 5 ($M = 31.88$, s.d. = 10.74), Experiment 6 ($M = 32.89$, s.d. = 10.81). Among participants who provided gender information, 931 identified as female, 957 as male, and 21 as other. Across experiments, the distribution was as follows: Experiment 1 (female = 135, male = 130, other = 5), Experiment 2 (female = 98, male = 44, other = 1), Experiment 3 (female = 94, male = 155, other = 6), Experiment 4 (female = 96, male = 55, other = 1), Experiment 5 (female = 181, male = 198, other = 3), Experiment 6 (female = 127, male = 174, other = 4). Participants received £2 for their participation, as well as a performance-based bonus of on average £0.25. Informed consent was obtained from all participants. At the end of the study, participants were fully debriefed about the experimental aims and informed that the social information had been generated by a computer algorithm. Ethics approval for Experiment 1 was obtained from the Ethics Committee of VU University Amsterdam (reference number: VCWE-2022-098), where author B.L. was based at the time. Experiments 2–6 were conducted while B.L and D.S were based at a Swedish institution. Under the Swedish Ethical Review Act (SFS 2003:460), this type of anonymous non-invasive research does not fall within the scope of studies requiring ethical review. As pre-registered for Experiment 1, we planned to exclude all participants who failed to respond on more than 20% of learning phase trials. However, preliminary analysis showed that this exclusion criterion had no qualitative effect. Hence, we did not exclude any participants.

### Experimental design
Across all experiments, the task was a binary (reward = 1, no reward = 0) bandit with probabilistic rewards (except for the social influence task in the test phase of Experiment 5, see Fig. 2). The task had two choice options, except for Experiment 4, which involved a four-choice bandit. Participants were randomly and blindly assigned to conditions, and choice option positions were randomized on each trial, except for Experiment 4. For Experiment 1, we conducted a priori simulation-based power analysis on the basis of a generative reinforcement learning model, indicating that 150 participants per group provide above 0.9 power to detect a difference between the groups in the first test-phase trial. Subsequent experiments follow this result. Performance was incentivized. The bonus was based on randomly selecting 6 outcomes (0 or 1). These outcomes were summed and divided by 10 as basis for the bonus (which consequently ranged from £0 to 0.6). In all experiments, participants could observe the aggregate social information produced by ostensible previous participants ('demonstrators'), who were described as performing the task individually (see Fig. 2). Although participants received the instruction that the demonstrators were past participants, demonstrators were in fact computer controlled and programmed to select each option with specific probabilities (see below). We sacrificed a deception-free design for the experimental control required to test our hypotheses. The tasks were administered online and were programmed using HTML, CSS and JavaScript within the platform psiTurk.

### Experiments 1 and 2
These nearly identical experiments differed only in the type of displayed social feature (see Fig. 2 and Supplementary Fig. 1). The experiments were divided into two phases: in the learning phase, participants completed 50 trials in which they selected between two options (we refer to these as X and Y) with fixed probabilities of reward ($P(\text{Reward}|X) = 0.8$, $P(\text{Reward}|Y) = 0.2$).

In Experiment 1 (pre-registered at https://aspredicted.org/H9P_YLD), the social information provided was the choices of 100 demonstrators. Participants were assigned to either of two experimental conditions, in which social information and rewards were either aligned (congruent condition) or misaligned (incongruent condition). In the congruent condition, each demonstrator selected option X with $P = 0.8$

(and option Y with $P = 0.2$). For the incongruent condition, this was reversed. Consequently, the environmental reward probabilities and the social information were aligned for participants in the congruent condition and misaligned for the incongruent condition.

Experiment 2 (pre-registered at https://aspredicted.org/HK1_BQQ) was equivalent to Experiment 1, but instead of the demonstrators' choices, participants were now provided with the demonstrators' mean reward in the trial. Thus, in the congruent condition, the mean observed reward for selecting cue X was 0.8 and for cue Y 0.2, and vice-versa in the incongruent condition (see Supplementary Fig. 1). No information about the number of demonstrators selecting each option was provided.

In the test phase (10 trials in Experiment 1, 20 trials in Experiment 2), participants chose among two new options (Z and W) with identical reward probability ($P = 0.5$). Again, the social information was aligned with one of the two options, following the same probability distribution as in the learning phase (0.8/0.2 choice preference [Experiment 1] or payoff distribution [Experiment 2]). This allowed us to estimate how experience in the learning context carried over to choices involving new cues (see Fig. 2). Choices in both the learning and test phases were incentivized.

### Experiment 3
Experiment 3 (pre-registered at https://aspredicted.org/DCD_YB3) mirrored Experiments 1 and 2 in its basic structure, but participants observed two (choice and payoff information) instead of one social feature in each trial. The position of the social feature was counterbalanced between participants (below or above the choice option). Depending on a between-subjects condition, one social feature was reward predictive (as in Experiments 1 and 2) and the other was random (that is, not predictive of reward). The reward-predictive feature had an average value of 0.8 (that is, on average 80% of others' choices were in favour of the high-reward option, or others' payoffs for the high-reward option were on average 0.8), while the random feature had a feature value drawn from a binomial distribution with mean of either $P = 0.8$ or $P = 0.2$, determined randomly on each trial. This design was employed to obtain a feature that was (1) uncorrelated with reward and (2) with values comparable to the reward-predictive feature (that is, equally far from 0.5).

To test our hypothesis that only the reward-predictive and not the random feature would guide choices, we constructed a test phase in which social features were opposed. Each choice option always had, on each trial, a high-payoff feature and a low-frequency feature, or a low-payoff feature and a high-frequency feature.

### Experiment 4
Experiment 4 (pre-registered at: https://aspredicted.org/PKH_DL1) followed the basic design of Experiment 1 but featured two additional choice options. On each trial, participants could choose between four options, one of which yielded rewards with $P(\text{reward}) = 0.8$ and three yielded rewards with $P(\text{reward}) = 0.2$. The two novel choice options were associated with intermediate social information: in the congruent condition, the four options were chosen by 65%, 15%, 15% and 5% of others. This pattern was reversed in the incongruent condition, such that the options were chosen by 5%, 15%, 15% and 65% of others. The addition of these intermediate choice options allowed participants to choose neither the majority nor the minority option. The test phase featured four novel choice options and was conceptually equivalent to Experiment 1.

### Experiment 5
Experiment 5 (not pre-registered) was identical to Experiment 1 with respect to the learning phase. However, we replaced the test phase with a standardized social influence task (the BEAST[65]). In each of the five trials, the participants' task was to estimate the number of

animals that were briefly presented (5 seconds) on the screen. Estimates were incentivized for accuracy using the same method as in ref. 65 (with the difference that an accurate estimate was worth £0.5). After the presentation period, participants submitted an initial estimate (Estimate 1). Subsequently, participants received social information in the form of the estimate of a previous participant who completed the task without social information. This estimate was selected conditional on the participants' estimate, to represent a 20% improvement (that is, decreased distance between Estimate 1 and the true number of animals). Next, participants could revise their estimate (Estimate 2). The change between Estimate 1 and Estimate 2 relative to the presented social information provides a measure of social influence $S_{\text{Beast}}$. $S_{\text{Beast}} = 1$ indicates that the social information completely replaced the individual estimate and $S_{\text{Beast}} = 0$ indicates that the social information did not affect the individual estimate. Values between 0 and 1 indicate a weighted average between individual and social information. Negative values mean that Estimate 2 moved away from the other participants' estimate. We removed data points where $|S_{\text{Beast}}| > 1$, but note that the results were qualitatively unchanged if including all data points, or only including data points where $0 \leq S_{\text{Beast}} \leq 1$.

### Experiment 6
In Experiment 6 (pre-registered at https://aspredicted.org/D6S_PMP), we used a within-participants design with two conditions (combined and single), followed by a transfer test (see Fig. 3). The order of the combined and single conditions was randomized across participants. The combined condition was equivalent to the congruent condition in Experiment 1 (see above). Again, participants made choices between two cues (A and B), where the A cue was favoured by the demonstrators (on average 80% selected the A cue). Selecting the A cue was rewarded with $P = 0.8$, while cue B was rewarded with $P = 0.2$. The single condition was structurally identical to the combined condition (participants chose between two new options C and D), with the key difference that no social feature was displayed (see Fig. 3).

In the incentivized transfer test, participants made choices between the A and C cues (high-value pair, that is, $P(R|A) = P(R|C) = 0.8$) and low-value B and D cues (low-value pair, $P(R|B) = P(R|D) = 0.2$), without social features present and without further reward feedback. This phase allowed us to test participants' preference for cues trained together with social features (A and B) versus cues trained individually (C and D).

### Statistical analysis
All analyses were conducted using the software R v.4.3.0. When analysing differences between experimental conditions at the first trial of the test phase, we used standard logistic regression. We verified that these models met the assumptions of binary and independent data points. We report the unstandardized estimates as effect size and Wald confidence intervals. For other comparisons, we used unpaired two-sample Wilcoxon tests with continuous responses. For linear regression, we used robust regression as implemented in the MASS package (v.7.3-65) in R. For analyses of repeated measures, we fitted generalized linear mixed models using the lme4 (ref. 101) package. These models included random intercepts and all relevant random slopes by participant and used a binomial error distribution (that is, logistic regression).

### Model fitting and comparison
Computational models were fit to individual participants' data using maximum likelihood methods. To avoid local minima, 20 different random starting points were used. Random effect model comparison was conducted on the basis of protected exceedance probabilities (pxp), which indicates the probability that one model is the most common in the population, corrected for chance. The individual Akaike information criteria (AIC) values were used as approximations to

the model evidence. See Supplementary Information Section 1 for details concerning the fixed heuristics and value shaping models. Because there were two versions of both the SFL and value shaping models (with and without a prior parameter), we first assessed, for each experiment, which version provided the best fit and used this version for the overall model comparison.

### Model recovery
Model identifiability analyses were conducted by simulating data from each model on the basis of the parameters of Experiment 1, and then fitting each model to each simulated dataset. If a model best fits the dataset it generated, it can be said to be identifiable. Supplementary Information Section 3 explains this approach and displays its results.

### Out-of-sample model prediction
Out-of-sample (or experiment) model predictive accuracy was assessed by calculating the log-likelihood for each participant in Experiments 2–6, on the basis of the median estimated parameters from Experiment 1 (see Supplementary Table 1). Protected exceedance probabilities and posterior model frequencies were calculated on the basis of the individual log-likelihoods. We did not penalize model complexity in this analysis since there was no model fitting.

### Agent-based model simulations
We conducted agent-based model (ABM) simulations to test whether social learning strategies emerge from the SFL model in various social learning scenarios. In these simulations, a population of 100 individual agents made choices in a two-armed bandit over 100 time steps (unless specified otherwise, as detailed below. See Supplementary Fig. 9 for simulations with up to 16 choice options). Each agent was controlled by the SFL model. For simulations involving the 'age' feature, each run consisted of 1,000 time steps. Each individual had a fixed probability of 0.01 to die in each time step, and dead individuals were replaced with naïve individuals. The SFL parameters were consistently set to $\alpha = 0.2$ and $\beta = 0.1$ for the results reported in the main text. See Supplementary Figs. 12 and 13 for simulations with alternative values for the learning parameters.

For each scenario and feature combination, we ran at least 500 independent simulation runs. All weights were initialized as 0. In each run, the reward probabilities associated with the bandit arms were individually randomized. The 'temporal variability' scenario manipulated the probability, per time step, that the reward probability of one arm would be re-randomized. The 'spatial variability' scenario introduced a per-time-step probability of migration between two groups. In this particular scenario, each run consisted of 200 time steps (except when involving the age feature, see above) to avoid confounding migration with less experience. In the 'dangerous environment' scenario, each simulation run spanned 1,000 time steps in an eight-armed bandit environment. Each individual had a fixed probability of 0.01 to die in each time step (in addition to individuals dying from selecting dangerous options). Each deceased individual was replaced with a new, naïve individual. In the 'competition' scenario, the reward for selecting action $i$, $R_i$, depended on the number of other individuals selecting the same action in that time step, following $R_i = \min\left[\frac{R}{n^c}, R\right]$, where parameter $c$ is the degree of competition and $R$ is the value of the option in the absence of competition. When $c = 0$, there is no competition, and $c = 1$ when there is full exploitative competition (that is, individuals receive $R = 1/n$)[76].

For simulations involving 'others' choices' and 'payoff' features, we employed the same feature representation in the ABM as when modelling the empirical data. The mean of the social feature vector was subtracted from each value, allowing feature weights to become negative (for example, to allow a minority preference). In simulations with 'age' and 'success' features, we utilized min–max normalization to accommodate the large range of potential values for these features. 'Age' corresponds to the lifetime of the individual at time $t$, while 'success' is its cumulative reward at time $t$. Consequently, the absolute magnitude of age and success feature weights are not directly comparable to others' choices and payoff feature weights.

### Reporting summary
Further information on research design is available in the Nature Portfolio Reporting Summary linked to this article.

## Data availability
The data are available at the Open Science Framework at https://osf.io/jry9x/ (ref. 102).

## Code availability
Model and analysis code is available at the Open Science Framework at https://osf.io/jry9x/ (ref. 102).

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

## Acknowledgements

We thank P. van den Berg, A. Bluet, W. van den Bos, C. Efferson, A. Gradassi, A. Olsson and I. Selbing for helpful comments and discussions. This work was funded by a Wallenberg Academy Fellow grant from the Knut and Alice Wallenberg Foundation (KAW 2021.0148) to B.L. and a Starting Grant (SOLAR ERC-2021-STG – 101042529) from the European Research Council to B.L. The funders had no role in study design, data collection and analysis, decision to publish or preparation of the manuscript.

## Author contributions

B.L., D.S. and L.M. conceived and designed the study. D.S. collected data. B.L. analysed data. B.L. performed computational modelling and agent-based simulation. B.L. and D.S. interpreted results. B.L., D.S. and L.M. prepared the manuscript, reviewed the results, edited the manuscript and approved the final version.

## Funding

## Competing interests

The authors declare no competing interests.

## Additional information

**Supplementary information** The online version
contains supplementary material available at

**Correspondence and requests for materials** should be addressed to
David Schultner or Björn Lindström.

**Peer review information** *Nature Human Behaviour* thanks
Wataru Toyokawa, Natalia Vélez and the other, anonymous,
reviewer(s) for their contribution to the peer review of this work.
Peer reviewer reports are available.

# Reporting Summary

## Statistics

For all statistical analyses, confirm that the following items are present in the figure legend, table legend, main text, or Methods section.

| n/a | Confirmed | |
|---|---|---|
| ☐ | ☒ | The exact sample size (*n*) for each experimental group/condition, given as a discrete number and unit of measurement |
| ☐ | ☒ | A statement on whether measurements were taken from distinct samples or whether the same sample was measured repeatedly |
| ☐ | ☒ | The statistical test(s) used AND whether they are one- or two-sided<br>*Only common tests should be described solely by name; describe more complex techniques in the Methods section.* |
| ☐ | ☒ | A description of all covariates tested |
| ☐ | ☒ | A description of any assumptions or corrections, such as tests of normality and adjustment for multiple comparisons |
| ☐ | ☒ | A full description of the statistical parameters including central tendency (e.g. means) or other basic estimates (e.g. regression coefficient) AND variation (e.g. standard deviation) or associated estimates of uncertainty (e.g. confidence intervals) |
| ☐ | ☒ | For null hypothesis testing, the test statistic (e.g. *F*, *t*, *r*) with confidence intervals, effect sizes, degrees of freedom and *P* value noted<br>*Give P values as exact values whenever suitable.* |
| ☒ | ☐ | For Bayesian analysis, information on the choice of priors and Markov chain Monte Carlo settings |
| ☐ | ☒ | For hierarchical and complex designs, identification of the appropriate level for tests and full reporting of outcomes |
| ☐ | ☒ | Estimates of effect sizes (e.g. Cohen's *d*, Pearson's *r*), indicating how they were calculated |

*Our web collection on statistics for biologists contains articles on many of the points above.*

## Software and code

Policy information about availability of computer code

| | |
|---|---|
| Data collection | The tasks were administered online and were programmed using the platform psiTurk (version 3.3.1) |
| Data analysis | We used R (version 4.3.0) and the lme4 package (version 1.1-35) to analyze the data. We used the MASS package (version 7.3-65) for robust regression. |

For manuscripts utilizing custom algorithms or software that are central to the research but not yet described in published literature, software must be made available to editors and reviewers. We strongly encourage code deposition in a community repository (e.g. GitHub). See the Nature Portfolio guidelines for submitting code & software for further information.

## Data

Policy information about availability of data

All manuscripts must include a data availability statement. This statement should provide the following information, where applicable:

- Accession codes, unique identifiers, or web links for publicly available datasets
- A description of any restrictions on data availability
- For clinical datasets or third party data, please ensure that the statement adheres to our policy

The data are available at the Open Science Framework: https://osf.io/jry9x/

# Research involving human participants, their data, or biological material

Policy information about studies with human participants or human data. See also policy information about sex, gender (identity/presentation), and sexual orientation and race, ethnicity and racism.

| | |
|---|---|
| Reporting on sex and gender | Participants self-reported gender. We did not perform any gender-based analysis, since our theoretical model did not make any predictions concerning gender-differences. Furthermore, reporting gender was not mandatory during participation, meaning that not all participants provided this data. |
| Reporting on race, ethnicity, or other socially relevant groupings | No reporting is made of race, ethnicity, or other socially relevant grouping variables. Our theoretical model made no apriori predictions about such variables. |
| Population characteristics | See above. |
| Recruitment | The participants were recruited on Prolific Academic. The study was described as a "decision-making experiment". Given this, we belive it is unlikely that self-seletion or related biases impacted the results. |
| Ethics oversight | Karolinska Institutet |

Note that full information on the approval of the study protocol must also be provided in the manuscript.

# Field-specific reporting

Please select the one below that is the best fit for your research. If you are not sure, read the appropriate sections before making your selection.

☐ Life sciences        ☒ Behavioural & social sciences        ☐ Ecological, evolutionary & environmental sciences

For a reference copy of the document with all sections, see nature.com/documents/nr-reporting-summary-flat.pdf

# Behavioural & social sciences study design

All studies must disclose on these points even when the disclosure is negative.

| | |
|---|---|
| Study description | In the basic experiment, participants learned which of two options (colored squares) yielded rewards through repeated choices. These two choice options were accompanied by social information, such as the ostensible choices of 100 previous participants, which were either positively or negatively associated with a participant's own reward. In this way, participants could learn to associate social information with own reward. In a subsequent test phase, participants faced two novel choice options, again presented alongside social information. We tested whether the learning phase shaped the test phase, such that participants were expected to be more likely to copy the majority if they had learned that majorities predict rewards, and more likely to copy the minority if they had learned that minorities predict rewards. Subsequent experiments varied this basic experimental design. |
| Research sample | The research sample consisted of participants recruited on Prolific Academic. Participants were required to be fluent in English, but not required to be native speakers. Their mean age was 33.8 years (sd = 11.96). All participants provided informed consent before starting the Experiment. Among participants who provided gender information, 931 identified as female, 957 as male, and 21 as other.<br>The sample was not representative. Because the purpose of the study was to test a theoretical model that in its current form does not make any predictions about participant characteristics, this sample was deemed appropriate. |
| Sampling strategy | We recruited participants who fulfilled the requirements outlined above at random from Prolific. Simulation-based power analysis was conducted for Experiments 1. The SFL model was simulated under random parameter values for a large number of repetitions . We assessed the % of runs where our planned analysis method provided a statistically significant result, given a true effect. For study 1, this indicated that 150 participants per group provided a power exceeding 0.9 of detecting a difference between the groups in the first Test phase trial. No power analysis was conducted for the following experiments but their sample sizes were chosen to be similar to Experiment 1 |
| Data collection | The tasks were administered online and were programmed using HTML, CSS and JavaScript within the platform psiTurk. Prolific Academic was used for recruitment. Conditions were automatically randomized without the researchers' involvement. |
| Timing | For Experiment 1, we started and stopped collecting data in October 2022. For Experiments 2 & 6, we started collecting data in March 2023 and stopped collecting data in April 2023. For Experiment 5, we started and stopped collecting data in June 2023. For Experiments 3 & 4, we started collecting data in September 2024 and stopped collecting data in October 2024. |
| Data exclusions | No data were excluded. |
| Non-participation | No participants dropped out. |
| Randomization | Allocation to experimental groups was random, and controlled by the experimental software. |

# Reporting for specific materials, systems and methods

We require information from authors about some types of materials, experimental systems and methods used in many studies. Here, indicate whether each material, system or method listed is relevant to your study. If you are not sure if a list item applies to your research, read the appropriate section before selecting a response.

## Materials & experimental systems

| n/a | Involved in the study |
|-----|----------------------|
| ☒ ☐ | Antibodies |
| ☒ ☐ | Eukaryotic cell lines |
| ☒ ☐ | Palaeontology and archaeology |
| ☒ ☐ | Animals and other organisms |
| ☒ ☐ | Clinical data |
| ☒ ☐ | Dual use research of concern |
| ☒ ☐ | Plants |

## Methods

| n/a | Involved in the study |
|-----|----------------------|
| ☒ ☐ | ChIP-seq |
| ☒ ☐ | Flow cytometry |
| ☒ ☐ | MRI-based neuroimaging |

## Plants

| | |
|---|---|
| Seed stocks | *Report on the source of all seed stocks or other plant material used. If applicable, state the seed stock centre and catalogue number. If plant specimens were collected from the field, describe the collection location, date and sampling procedures.* |
| Novel plant genotypes | *Describe the methods by which all novel plant genotypes were produced. This includes those generated by transgenic approaches, gene editing, chemical/radiation-based mutagenesis and hybridization. For transgenic lines, describe the transformation method, the number of independent lines analyzed and the generation upon which experiments were performed. For gene-edited lines, describe the editor used, the endogenous sequence targeted for editing, the targeting guide RNA sequence (if applicable) and how the editor was applied.* |
| Authentication | *Describe any authentication procedures for each seed stock used or novel genotype generated. Describe any experiments used to assess the effect of a mutation and, where applicable, how potential secondary effects (e.g. second site T-DNA insertions, mosiacism, off-target gene editing) were examined.* |

