## [Peer Review File · Nature Human Behaviour]

Feature-based reward learning shapes human social learning strategies

Corresponding Author: Dr Björn Lindström

A version of this paper was originally rejected for publication by Nature Human Behaviour, however that decision was reconsidered after appeal by the authors.

Version 0:

Decision Letter:

7th August 2024

Dear Dr Lindström,

Thank you once again for your manuscript, entitled "Feature-based reward learning shapes human social learning strategies," and for your patience during the peer review process.

Your manuscript has now been evaluated by 3 reviewers, whose comments are included at the end of this letter. In the light of their advice, I regret that we cannot offer to publish your manuscript in Nature Human Behaviour.

While the reviewers find your work of some interest, they raise concerns about the strength of conclusions that can be drawn from this study and the appropriateness of the technical approach. We feel that these reservations are sufficiently important as to preclude publication of this work in Nature Human Behaviour.

I am sorry that we cannot be more positive on this occasion but hope that you will find our reviewers' comments helpful when preparing your paper for submission elsewhere.

Sincerely,

Nature Human Behaviour

Reviewer expertise:

Reviewer #1: Reinforcement learning, Social learning

Reviewer #2: Social learning

Reviewer #3: Reinforcement learning

Reviewers' Comments:

Reviewer #1:

Remarks to the Author:

OVERVIEW

Social learning is adaptive because it is selective. Prior work has documented a range of social learning strategies that humans use to decide when to learn from others and from whom to learn, including "copy when uncertain" and "copy the majority". These strategies are often characterized as a collection of heuristics—however, this heuristics account alone does not explain why there are inter-individual differences in which strategies people rely on, or how people resolve conflicts when two heuristics favor different choices. Here, the authors introduce a social feature learning framework. The key claim advanced in this paper is that this domain-general framework provides a parsimonious explanation of social learning strategies and can also explain additional

phenomena such as inter-individual variability. As evidence for this claim, the authors present four studies where participants combine asocial cues and different social cues with varying levels of reliability to make decisions. They then use a series of model simulations to establish that social feature learning can give rise to a variety of social learning strategies and resolve conflicts between them.

There is a lot to be excited about in this paper. This paper provides strong empirical support to a longstanding conjecture in the literature—that social learning strategies are governed by domain-general, evolutionarily-conserved learning mechanisms. However, I am not sure how to evaluate the scope of the claim being made in the paper. Is the SFL framework intended to describe a finite set of social learning strategies under a common framework, or is it a more general account of how people learn to learn from social information from experience? I explain this point, and other, minor suggestions, in more detail below.

MAJOR SUGGESTIONS:

(1) The central claim of the paper is that social learning strategies are governed by a common, domain-general learning mechanism. I am not sure how broad of a claim the authors are trying to make here—is the goal of this framework to describe a finite set of social learning strategies under a common framework, or to provide a more general account of how social learning is shaped by experience?

If the authors are trying to make the latter, stronger claim, then there are several important social learning phenomena that do not seem to be adequately explained by the SFL framework. Prior work using associationist tasks—such as the setup used here—have documented asymmetries between how people learn from social and asocial information. For example, people are curiously optimistic about the quality of social information in a way that's consistent with a biased updating mechanism (Leong & Zaki, 2018), and they more flexibly update beliefs about bad people compared to good people (Siegel et al., 2018). These biased updating mechanisms do not seem to have a clear analogue to asocial learning—can they be explained under a common, domain-general learning mechanism?

If, instead, the authors are trying to make the former, more focused claim, it would be helpful to state more precisely what phenomena this framework captures. For example, the behavioral and simulation results provide evidence that the SFL captures frequency-dependent copying, payoff-biased copying, prestige bias, copying from older individuals, etc., and it could feasibly capture other social learning strategies, such as size-, sex-, dominance-biased copying. To be clear, even this more focused claim would be a great contribution, and readers like me would not be stuck wondering why particular social learning phenomena (e.g., inferential social learning, asymmetric updating mechanisms, etc.) seem to be overlooked.

(2) Whichever version of this claim the authors are trying to make, I was surprised to see that metacognitive social learning strategies were not mentioned in this paper. Prior associationist accounts of social learning strategies (e.g., Heyes, 2016) have proposed that—while most SLSs are governed by domain-general processes—learning from knowledgeable others presents an exception and may be governed by separate processes. Where do these social learning strategies fall within the SFL framework?

(3) In my opinion, the claim that social feature weights are correlated with later social influence scores is not well supported by the evidence (Experiment 3). Based on Figure S2, it seems that the distribution of social influence by social feature weights is uniform, and that this correlation may be largely driven by a handful of points in the $w_{\text{others}} = [-1, -0.33]$ range. How robust is this result to outliers?

MINOR SUGGESTIONS:

(1) I think that there is not enough methodological detail in the main text for the behavioral results to stand alone. For example, I had to dive into the methods at the very end of the paper to interpret Figure 2. I assumed, from the figure caption, that the Training phase plots are on the left and the Test phase plots are on the right, but this is never mentioned explicitly, and the gray shaded areas at the start of the test phase are not explained. It was also not obvious to me why the lines in the Test phase converge to the middle – readers have to dig deep into the methods to learn that the two new cues introduced in the Test phase have a 0.5 reward probability. Consider adding a high-level summary of the experimental task (1 paragraph) in the main text and adding additional annotations to your figures and detail to the figure captions.

(2) Minor typos:
p.32: "Experiential design"

Reviewer #2:

Remarks to the Author:

Dear Editor,

Summary of the Paper:

The authors examined human social learning strategies through a domain-general reinforcement learning model called Social Feature Learning (SFL). They propose that individuals can learn which features of social information (such as social frequency and others' payoffs) predict rewards and update the weight they give to social information in the reinforcement algorithm accordingly. They demonstrate that this reward learning approach can parsimoniously generate a range of different social learning strategies, such as copy-the-majority and copy-the-successful behaviors, depending on the individual's past reward experiences. The authors validate the SFL model using a series of human behavioral experiments, confirming that SFL provides a better fit to human data compared to other candidate models, including value-shaping (where social cues act as a bonus reward) and decision-biasing models.

Experiment 4, which tested whether learning both non-social and social features would result in Feature Competition—a well-known phenomenon in associative learning—was particularly compelling. The results confirmed that human value learning is impaired when learning from both social and non-social cues, suggesting that the same learning mechanism operates in both non-social value learning and social learning. The post-hoc agent-based simulation of SFL suggests that previous social learning theories and empirical findings might be unified by this domain-general associative learning mechanism.

The manuscript is well-written, and the topic is of great interest to a wide range of scientific disciplines, not only in human behavioral sciences but also in evolutionary ecology. However, I have significant concerns regarding the analyses and interpretations provided. My main concerns stem from two primary points. First, the exploration of free parameters (such as alpha and beta) was very limited. Second, the model simulations and experiments were mostly restricted to two-armed bandits (except for the dangerous environment manipulation). These limitations narrow the reach of their findings, raising doubts about whether their conclusions can be generalized to more realistic scenarios with a greater number of behavioral options. I elaborate on these two major points below:

(1) A Wider Range of Parameter Values in Simulations and Reporting Empirical Fit Values

In the a priori model prediction simulations, the authors used a uniform distribution where $\alpha = U(0.001, 0.4)$ and $\beta = U(0.0001, 0.2)$, despite alpha ranging from 0 to 1 and beta from 0 to $+\infty$. This limited range of parameter values may have biased their predictions. For instance, a larger alpha (e.g., $\alpha = 0.6$; myopic learning) combined with a small beta is known to generate risk aversion (the "hot stove effect"). Additionally, a limited beta makes agents very exploitative. In the post-hoc agent-based model, the parameters were fixed at $\alpha = 0.2$ and $\beta = 0.1$, which could also bias the results. The authors should report results for a wider range of parameter values to cover the behavior of both myopic learners and explorative decision-makers.

Furthermore, I am curious about the experimentally calibrated parameter values. Reporting the parameter fit results would enhance the robustness of their findings.

(2) Dual Alpha Model?

Related to the point above, I wonder why the authors did not test a model with two alphas (one for non-social feature updating and one for social feature updating). Testing such a dual-alpha model with human data, and ideally demonstrating that the single-alpha model is the winning model, would provide stronger evidence for the main finding that the same learning mechanism (with common alpha values) underlies both social and non-social feature learning. Currently, it is unclear if a single alpha suffices or if this result is due to the lack of consideration of multi-alpha models.

(3) Multi-Armed Tasks?

Studying conformity/anti-conformity bias using a two-armed bandit could mislead conclusions because avoiding the majority always means following the minority. Previous literature has provided reasons why focusing solely on two-option tasks is inadequate (Muthukrishna et al., 2016). The authors should explore tasks with a greater number of options (such as the four-armed or eight-armed bandits used in the dangerous environment simulation).

Muthukrishna, M., Morgan, T. J., & Henrich, J. (2016). The when and who of social learning and conformist transmission. *Evolution and Human Behavior*, 37(1), 10-20.

Minor points:

(4) What was the non-social features?

They used a cover-story of "rabbit hunt" where the non-social feature was the size of the rabbit shown (s_{rabbit} in Eq. 1). However, according to the supplementary Fig. S1, the task looked just like a 2-armed bandit task. How exactly did they manipulate the non-social feature? The authors should explain it in the main text (maybe soon after the Eq. 1 where they showed an example of $s_{\text{others}} = 0.8$).

(5) Decision-biasing model

Their "fixed heuristic" model is also called "decision-biasing" in the literature. Perhaps it would help readers connect the current work with those previous works if the authors mention it when they introduce the term. (and the ref. 23 is also using the similar decision-biasing model though the authors skip citing it there).

(6) Figure 6: why do the positions of initial points differ between panels?

The distribution of points at $x = 0$ differs between panels (except for the panel a and d, they are quite close). My understanding is that the focal feature was switched off when $x = 0$, which should make the balance between the four weights always the same among those panels. Could the authors clarify why the initial points differ?

(7) Why did w_{others} not become negative?

In the competition manipulation (fig. 6d), w_{others} seems to converge to zero rather than a negative value. If copying "minority" helps avoid scramble competition, why does w_{other} not become negative?

(8) Minor typo? Fig. 7 legend: "while the dotted horizontal line ..." should be "the dashed diagonal line" instead?

Reviewer #3:

Remarks to the Author:

Review for social feature learning by

This study (by Schultner et al) proposes a reward learning framework to explain the emergence of social learning strategies. Through 4 experiments and multi-agent simulations they show that people adjust their learning based on rewards associated with social features like behavior and success and number of people ("majority"). There is much to like in this paper, starting from the fundamental question, and passing through the exaction and the open-science practices adopted by the authors.

I do have, however, some questions, suggestions and remarks to share with the authors and the editors.

1/ First, I believe that a key experimental test of their theory would consist of a task where multiple features (i.e., number of subjects and payoff are simultaneously displayed to the participants, ideally in two versions (either payoff is predictive or number of subjects is predictive) to show that they model work in the more ecological situations where multiple features are presented. I believe this is partially addressed by the multi-agent simulations (that show that the models can work, in theory) but not empirically (in all experiments there is only one social feature). I am also aware that the authors already produced a lot for work, so I will defer to the editors' opinion concerning how important this additional experiment could be.

2/ The authors mainly focus on the simulations of their model. However, as readers, we are left without information concerning whether or not the other models were able to reproduce the same behavioural signatures (model falsification: see Palminteri et al. 2027) (I suspect that, to some extent, yes, especially the heuristics models - but this would not be the case in the 'two social features experiment " suggested above). Similarly, what is the model recovery/confusion matrix?

3/ I think that the authors are missing a very relevant literature here, specifically concerning Rieskamp's work on strategy selection: <https://psycnet.apa.org/buy/2006-06642-005>
some of these studies propose precise models about how associative learning mechanisms may explain how heuristics are selected. Of course, there are many differences, but still I believe this literature is relevant.

4/ I wonder how much the authors really commit to the idea that I will summarize in this sentence: "There are no hard-coded social learning heuristics, they are all learned by trial-and-error from rewards". This is probably an extreme, unnuanced version but the authors seem nonetheless seem to lean toward this view. Now, the problem here is that I do not believe that this is fully true. We all have the intuition that social learning is automatically deployed before any reward is experienced (i.e., in childhood) and in many situations in which social learning, imitation, is used rewards are delayed. Can the authors comment on this and discuss these points?

5/ Finally, I believe that the Figures could be improved. Figure 1 could be a bit clearer (what the grey vs. white mean? the legend mentions rectangles, but there is no rectangle, sensu strictu, in the figure). Maybe adding the equations? Figure 2 some of the axes are without legend (maybe because it is the same as on the right column). Plus $p(\text{choice})$ is not very clear. More generally they could be made more informative.

6/ Was social information simulated, was this told to the participants or did the study involve actual social information?

**Following suitable revisions, you may want to consider transferring your manuscript. Although we cannot offer to publish your manuscript, we believe the Editorial Board at Scientific Reports will find it interesting and recommend you transfer there. To transfer your manuscript there, please use our manuscript transfer portal. You will not have to re-supply manuscript metadata and files, unless you wish to make modifications. For more information, please see our http://www.nature.com/authors/author_resources/transfer_manuscripts.html?
WT.mc_id=EMI_NPG_1511_AUTHORTRANSF&WT.ec_id=AUTHOR">WT.mc_id=EMI_NPG_1511_AUTHORTRANSF&WT.ec_id=AUTHOR manuscript transfer FAQ page.

Version 1:

Decision Letter:

Dear Dr Lindström,

Thank you for your correspondence asking us to reconsider our decision on your Article, "Feature-based reward learning shapes human social learning strategies". After careful consideration we have decided that we would be willing to consider a revised version of your manuscript.

In your revision, please incorporate the two additional experiments, the agent-based simulations, as well as the control analyses and sensitivity check you have conducted in response to the reviewers' concerns. In addition, please also address the conceptual concerns raised by Reviewer #1.

Along with your revised manuscript, you should also submit a separate point-by-point response to all of the concerns raised by the referees, in each case describing what changes have been made to the manuscript or, alternatively, if no action has been taken, providing a compelling argument for why that is the case. If we feel that a substantial attempt has been made to address the referees' comments, this response will be sent back to the referees - along with the revised manuscript - so that they can judge whether their concerns have been addressed satisfactorily or otherwise.

I should stress, however, that we would be reluctant to trouble our referees again unless we thought that their comments had been addressed in full.

- ensure it complies with our format requirements as set out in our [Guide to Authors](http://www.nature.com/nathumbehav/info/gta).

- state in a cover note the length of the text, methods and figure legends; the number of references and the number of display items.

Please ensure that all correspondence is marked with your Nature Human Behaviour reference number in the subject line.

Please use the following link to submit your revised manuscript:

Link Redacted

We hope to receive your revised paper within four weeks. If you cannot send it within this time, please let us know so that we can close your file. In this event, we will still be happy to reconsider your paper at a later date so long as nothing similar has been accepted for publication at Nature Human Behaviour or published elsewhere in the meantime. Should you miss the four-week deadline and your paper is eventually published, the received date will be that of the revised, not the original, version.

I look forward to hearing from you soon.

Best regards,

Nature Human Behaviour

Version 2:

Decision Letter:

Our ref: NATHUMBEHAV-24062301B

13th March 2025

Dear Dr. Lindström,

Thank you for submitting your revised manuscript "Feature-based reward learning shapes human social learning strategies" (NATHUMBEHAV-24062301B). It has now been seen by the original referees and their comments are below. As you can see, the reviewers find that the paper has improved in revision. We will therefore be happy in principle to publish it in Nature Human Behaviour, pending minor revisions to satisfy the referees' final requests and to comply with our editorial and formatting guidelines.

We are now performing detailed checks on your paper and will send you a checklist detailing our editorial and formatting requirements within two weeks. Please do not upload the final materials and make any revisions until you receive this additional information from us.

Sincerely,

Nature Human Behaviour

Reviewer #1 (Remarks to the Author):

I thank the authors for their thorough and thoughtful revision. The changes introduced in this revision have indeed strengthened the manuscript; I particularly appreciate the authors' efforts to clarify their theoretical position in the paper, conduct additional tests of their theory, and deepen their exposition of the modeling results.

I have one minor suggestion about a new passage introduced in the revision:

"In our experiments, participants quickly adapted their social learning strategies to changes in the reward associations of social features. Outside of the lab, this flexibility is likely to produce significant variability within and between individuals in what social learning strategies they use, and their reliance on social versus individual forms of learning. For example, individuals may develop and adopt a wide range of social learning strategies based on their unique experiences and the specific rewards they encounter."

This is an interesting discussion point, but I think the unique contribution made by this paper is unclear—the authors already cite empirical evidence for individual variability in social learning strategies elsewhere in the paper (e.g., Mesoudi et al., 2016). In the context of this prior work, I think the main contribution here is that the authors provide computational evidence that this variability can arise from a domain-general learning mechanism. It would be helpful to focus the claims made in this passage of the discussion.

Reviewer #2 (Remarks to the Author):

Dear Editor,

The authors revised the MS thoroughly, with all the questions and concerns raised by me and two other reviewers being addressed appropriately and compellingly.

I believe the MS is ready to be published as it is. I would like to say the warmest congratulations to the authors.

All the best,
Wataru Toyokawa

Reviewer #3 (Remarks to the Author):

I thank the authors for considering my points in a very constructive manner. I believe the new experiment is a valuable addition to an already solid paper. I also appreciate the results and the transparency regarding recovery, as well as the engagement in out-of-sample validation and falsification. I have no further comments.

Version 3:

Decision Letter:

Dear Dr Lindström,

We are pleased to inform you that your Article "Feature-based reward learning shapes human social learning strategies", has now been accepted for publication in Nature Human Behaviour.

With best regards,

Nature Human Behaviour

P.S. Click on the following link if you would like to recommend Nature Human Behaviour to your librarian <http://www.nature.com/subscriptions/recommend.html#forms>

** Visit the Springer Nature Editorial and Publishing website at http://editorial-jobs.springernature.com?utm_source=ejp_NHumB_email&utm_medium=ejp_NHumB_email&utm_campaign=ejp_NHumB for more information about our career opportunities. If you have any questions please click [here](mailto:editorial.publishing.jobs@springernature.com).

use, sharing, adaptation, distribution and reproduction in any medium or format, as long as you give appropriate credit to the original author(s) and the source, provide a link to the Creative Commons license, and indicate if changes were made. In cases where reviewers are anonymous, credit should be given to 'Anonymous Referee' and the source. The images or other third party material in this Peer Review File are included in the article's Creative Commons license, unless indicated otherwise in a credit line to the material. If material is not included in the article's Creative Commons license and your intended use is not permitted by statutory regulation or exceeds the permitted use, you will need to obtain permission directly from the copyright holder. To view a copy of this license, visit <https://creativecommons.org/licenses/by/4.0/>

Response to reviewers

For the manuscript “*Feature-based reward learning shapes human social learning strategies*” by David Schultner, Lucas Molleman, and Björn Lindström.

We thank the reviewers for their positive evaluations of our work, and their generous and constructive comments. In response, we have revised our manuscript to address all of their concerns and accommodate their suggestions. The most substantial revisions include (i) the addition of the two behavioural experiments (total $n=715$) suggested by the reviewers, the results of which closely match our model predictions, providing strong further support for our reward-learning account of social learning; (ii) presenting an extensive analysis that quantifies our model’s performance relative to alternative models, enhancing the embedding of our work in the relevant literature; and (iii) elaborations on our model’s limitations, duly qualifying the scope of our claims. We believe that our manuscript has substantially improved, and we hope the reviewers share our view.

Below we show the reviewer comments in *italics*, with our point-by-point response in normal font.

Reviewer #1:

Remarks to the Author:

OVERVIEW

Social learning is adaptive because it is selective. Prior work has documented a range of social learning strategies that humans use to decide when to learn from others and from whom to learn, including “copy when uncertain” and “copy the majority”. These strategies are often characterized as a collection of heuristics — however, this heuristics account alone does not explain why there are inter-individual differences in which strategies people rely on, or how people resolve conflicts when two heuristics favor different choices. Here, the authors introduce a social feature learning framework. The key claim advanced in this paper is that this domain-general framework provides a parsimonious explanation of social learning strategies and can also explain additional phenomena such as inter-individual variability. As evidence for this claim, the authors present four studies where participants combine asocial cues and different social cues with varying levels of reliability to make decisions. They then use a series of model simulations to establish that social feature learning can give rise to a variety of social learning strategies and resolve conflicts between them.

There is a lot to be excited about in this paper. This paper provides strong empirical support to a longstanding conjecture in the literature—that social learning strategies are governed by domain-general, evolutionarily-conserved learning mechanisms. However, I am not sure how to evaluate the scope of the claim being made in the paper. Is the SFL framework intended to describe a finite set of social learning strategies under a common framework, or is it a more general account of how people learn to learn from social information from experience? I explain this point, and other, minor suggestions, in more detail below.

Response: We thank the reviewer for their positive evaluation of our manuscript, and their constructive comments. We feel that addressing them has clarified our paper and substantially improved the presentation of our work.

MAJOR SUGGESTIONS:

(1) The central claim of the paper is that social learning strategies are governed by a common, domain-general learning mechanism. I am not sure how broad of a claim the authors are trying to make here—is the goal of this framework to describe a finite set of social learning strategies under a common framework, or to provide a more general account of how social learning is shaped by experience?

If the authors are trying to make the latter, stronger claim, then there are several important social learning phenomena that do not seem to be adequately explained by the SFL framework. Prior work using associationist tasks—such as the setup used here—have documented asymmetries between how people learn from social and asocial information. For example, people are curiously optimistic about the quality of social information in a way that’s consistent with a biased updating mechanism (Leong & Zaki, 2018), and they more flexibly update beliefs about bad people compared to good people (Siegel et al., 2018). These biased updating mechanisms do not seem to have a clear analogue to asocial learning—can they be explained under a common, domain-general learning mechanism?

If, instead, the authors are trying to make the former, more focused claim, it would be helpful to state more precisely what phenomena this framework captures. For example, the behavioral and simulation results provide evidence that the SFL captures frequency-dependent copying, payoff-biased copying, prestige bias, copying from older individuals, etc., and it could feasibly capture other social learning strategies, such as size-, sex-, dominance-biased copying. To be clear, even this more focused claim would be a great contribution, and readers like me would not be stuck wondering why particular social learning phenomena (e.g., inferential social learning, asymmetric updating mechanisms, etc.) seem to be overlooked.

Response: We thank the reviewer for this comment. We agree that it is very important to indicate the limitations of our model. While we believe that our model lays the groundwork for a truly general account of social learning, our paper provides evidence for the more focused claim about social learning strategies, that is, ways in which individuals selectively use social cues in decision-making (cf. Laland et al 2004). We rewrote the discussion section to reflect this point.

The relevant section (p. 31-32) now reads:

“There are several limitations to highlight. Firstly, the SFL model focuses on how individuals learn to selectively use observable (social) cues to explain the emergence of strategic social learning. This means that the model does not directly address inferential forms of social learning, which are important for teaching (Gweon, 2021; Vélez & Gweon, 2021). Augmenting the SFL model with approaches such as inverse reinforcement learning (Arora & Doshi, 2019), which can be used to infer the preferences or beliefs of others (Jara-Ettinger, 2019), could help address this limitation. Furthermore, in our implementation of the SFL model,

we used simple feature representations. However, real-world feature representations are typically multi-dimensional (Wise et al., 2024), attention-dependent (Niv et al., 2015), and often abstract (Radulescu et al., 2021), and updating of feature weights can be biased or moderated by the values of (other) features (FeldmanHall & Nassar, 2021; Leong & Zaki, 2018; Siegel et al., 2018). While it is possible to enhance the SFL model by incorporating more sophisticated feature representations (e.g., with deep neural networks, Mondragón et al., 2017) and asymmetric weight updating, the simplicity of linear features and unbiased updating enabled us to highlight the fundamental mechanistic principles of the model without sacrificing generality. It is also worth noting that the simple linear approximation sufficed for capturing human behaviour in our experiments (cf. Fig. 2), indicating that more complex representations may not always be advantageous.

Additionally, our study employed simplified, computerised experiments to study human social learning. Although this allowed us to design highly controlled settings where our theoretical model made distinct predictions (Kendal et al., 2018), future research should further test the SFL model using non-experimental, naturalistic data. Furthermore, our agent-based simulation setting, while comprehensive, did not cover the entire mosaic of social learning strategies. Future research could extend the SFL model to investigate how other important strategies, including preferences for familiar, prestigious or dominant others, can emerge through learning (Jiménez & Mesoudi, 2019). One interesting direction would be to explore the implications of individuals being able to explicitly represent feature weights, and communicate them to others, allowing for the “social learning of social learning” (Heyes, 2016; Mesoudi et al., 2016). Moreover, although the experiments conducted here test scenarios with immediate feedback, the SFL model can naturally be extended to scenarios with delayed rewards: Employing a temporal-difference updating rule (Sutton & Barto, 1998) would allow social feature learning even when rewards are delayed by sequences of actions or by the passing of time. Finally, while our theoretical agent-based model simulations demonstrate that reward learning is sufficient for social learning strategies to emerge in a population of naive individuals, confirmatory empirical evidence would require intensive developmental studies of the relationship between social information and rewards in children’s everyday environments. In line with the reward learning account, existing studies suggest that reward learning plays an important role in the development of imitation skills among children (Ray & Heyes, 2011). However, more developmental work of human social learning would be needed to elucidate how social learning strategies are shaped by everyday experience.”

(2) Whichever version of this claim the authors are trying to make, I was surprised to see that metacognitive social learning strategies were not mentioned in this paper. Prior associationist accounts of social learning strategies (e.g., Heyes, 2016) have proposed that—while most SLSs are governed by domain-general processes—learning from knowledgeable others presents an exception and may be governed by separate processes. Where do these social learning strategies fall within the SFL framework?

Response: Metacognitive social learning strategies (Heyes, 2016) involve individuals having representations of who to learn from, and when to do so. In our model, individuals do not necessarily have explicit, reportable representations of their social learning strategies. Rather, individuals behave in ways consistent with various social learning strategies based

on weights they assign to (social) features and decide depending on the feature values and their weights. In our paper, we consider situations where these weights are updated based on experiencing rewards in situations characterised by these features. However, there are situations in which people might be able to report (proximates of) such weights (e.g., “When I choose which restaurant I want to go to, I mostly base my decision on where the most people are, rather than how many stars they have on Google reviews.”). When made explicit, feature weights might be available for social transmission, enabling the “social learning of social learning” (Mesoudi et al., 2016). Future work based on our model might address this possibility. We now address this point in the Discussion (p. 32):

“One interesting direction would be to explore the implications of individuals being able to explicitly represent feature weights, and communicate them to others, allowing for the “social learning of social learning” (Heyes, 2016; Mesoudi et al., 2016).”

(3) In my opinion, the claim that social feature weights are correlated with later social influence scores is not well supported by the evidence (Experiment 3). Based on Figure S2, it seems that the distribution of social influence by social feature weights is uniform, and that this correlation may be largely driven by a handful of points in the $w_{\text{others}} = [-1, -0.33]$ range. How robust is this result to outliers?

Response: We thank the reviewer for the chance to provide additional details on this analysis. First, we used a robust regression model (rlm function in the MASS package), which is less sensitive to outliers than OLS regression. We have clarified when reporting the results of this analysis on p. 15 and in the methods section (p. 38).

However, to fully ascertain that this analysis is robust to the outliers the reviewer points out, we reran the robust regression analysis removing data points with $w_{\text{others}} < -0.33$, as suggested by the reviewer. The result of this regression was similar to the original analysis: $B = 0.21$, $SE = 0.09$, $t = 2.4$, $p = 0.017$. We found a similar relationship with OLS regression (with possible outliers removed: $B = 0.25$, $SE = 0.09$, $t = 2.85$, $p = .005$. Including possible outliers: $B = 0.21$, $SE = 0.7$, $t = 2.85$, $p = .005$). These control analyses are now included in “Section 2: Additional Analyses” in the SI.

We conclude that the relationship between the social feature weight and social influence observed in Experiment 5 (previously Experiment 3) is robust to outliers.

MINOR SUGGESTIONS:

(1) I think that there is not enough methodological detail in the main text for the behavioral results to stand alone. For example, I had to dive into the methods at the very end of the paper to interpret Figure 2. I assumed, from the figure caption, that the Training phase plots are on the left and the Test phase plots are on the right, but this is never mentioned explicitly, and the gray shaded areas at the start of the test phase are not explained. It was also not obvious to me why the lines in the Test phase converge to the middle – readers have to dig deep into the methods to learn that the two new cues introduced in the Test phase have a 0.5 reward probability. Consider adding a high-level summary of the

experimental task (1 paragraph) in the main text and adding additional annotations to your figures and detail to the figure captions.

Response: We thank the reviewer for pointing this issue out. In our updated manuscript we have clarified Fig. 2 (included below as Fig. R1-1) as well as its figure caption through the following changes:

1. We have modified panel *a* intending to give a more intuitive depiction of the experimental paradigm
2. We have added unambiguous 'Learning' and 'Test' panel headers.
3. The figure caption now describes that the gray shaded areas ("boxes") indicate the first trial of the Test phase, which amounts to the test case of our predictions.
5. We have added a high-level summary of the experimental task, including a description of the test phase reward probabilities in the section 'Empirical tests of the Social Feature Learning model' so that it is immediately clear that both choice options have equal reward probability: 'In the subsequent Test phase, participants faced new choice options with equal reward probability'

Fig. R1-1 | Social learning is shaped by individual reward experience. **a**, Participants completed a probabilistic decision-making task. Choice options had non-social (square colour) and social features (Exp. 1: Others' choices (as illustrated in **a**); Exp. 2: Others' payoffs). During the Learning phase, participants could learn which of two options was most rewarding. Social features and rewards were aligned in the Congruent condition and misaligned in the Incongruent condition. The subsequent Test phase was used to assess whether social learning was shaped by rewards; participants were exposed to two new options (different coloured squares). **b**, SFL model predictions (identical for the two social features, see Methods) of how reward experience in the Learning phase shapes social learning in the Test phase (faint lines: individual simulation runs; solid lines: averages; see Methods for details and Fig. S2 for simulations across a range of parameters). The model includes a weight prior parameter, which allows for a majority preference on the first trial of the Learning phase. **c-d**, Empirical results closely align with the SFL model; with lines showing averages with 95% CIs. **c**, In Experiment 1, participants learned to follow the majority (minority) in the Congruent (Incongruent) condition in the Learning phase. **d**, In Experiment 2, participants learned to follow (avoid) the option with high payoffs for others in the Congruent (Incongruent) condition

in the Learning phase. For both Experiments, social learning spilled over into the Test phase. Grey boxes highlight predictions at the outset of the Test phase; *** indicates $p < .001$.

(2) *Minor typos:*

p.32: “*Experiential design*”

Response: Fixed, thank you for spotting this.

Reviewer #2:

Remarks to the Author:

Dear Editor,

Summary of the Paper:

The authors examined human social learning strategies through a domain-general reinforcement learning model called Social Feature Learning (SFL). They propose that individuals can learn which features of social information (such as social frequency and others' payoffs) predict rewards and update the weight they give to social information in the reinforcement algorithm accordingly. They demonstrate that this reward learning approach can parsimoniously generate a range of different social learning strategies, such as copy-the-majority and copy-the-successful behaviors, depending on the individual's past reward experiences. The authors validate the SFL model using a series of human behavioral experiments, confirming that SFL provides a better fit to human data compared to other candidate models, including value-shaping (where social cues act as a bonus reward) and decision-biasing models.

Experiment 4, which tested whether learning both non-social and social features would result in Feature Competition—a well-known phenomenon in associative learning—was particularly compelling. The results confirmed that human value learning is impaired when learning from both social and non-social cues, suggesting that the same learning mechanism operates in both non-social value learning and social learning. The post-hoc agent-based simulation of SFL suggests that previous social learning theories and empirical findings might be unified by this domain-general associative learning mechanism.

The manuscript is well-written, and the topic is of great interest to a wide range of scientific disciplines, not only in human behavioral sciences but also in evolutionary ecology. However, I have significant concerns regarding the analyses and interpretations provided. My main concerns stem from two primary points. First, the exploration of free parameters (such as alpha and beta) was very limited. Second, the model simulations and experiments were mostly restricted to two-armed bandits (except for the dangerous environment manipulation). These limitations narrow the reach of their findings, raising doubts about whether their conclusions can be generalized to more realistic scenarios with a greater number of behavioral options. I elaborate on these two major points below:

Response: We thank the reviewer for their positive evaluation, and their thoughtful and constructive suggestions for improvement of our paper.

Following the reviewer's suggestions, we have conducted explorations of both the SFL model predictions for our experiments and the agent-based simulation under a wider variety of parameter settings. We have also conducted a new decision-making experiment with a 4-armed bandit ($n=353$), and conducted additional simulations with up to 16 arms. The results fully confirm the robustness of our original conclusions. We detail these analyses below.

(1) A Wider Range of Parameter Values in Simulations and Reporting Empirical Fit Values

In the a priori model prediction simulations, the authors used a uniform distribution where $\alpha = U(0.001, 0.4)$ and $\beta = U(0.0001, 0.2)$, despite α ranging from 0 to 1 and β from 0 to $+\infty$. This limited range of parameter values may have biased their predictions. For instance, a larger α (e.g., $\alpha = 0.6$; myopic learning) combined with a small β is known to generate risk aversion (the "hot stove effect"). Additionally, a limited β makes agents very exploitative.

Response: The reviewer is right that the simulation parameters in our original submission were limited (learning rate $\alpha = U(0.001, 0.4)$, softmax temperature $\beta = U(0.0001, 0.2)$). Our reason for limiting especially the β parameter to relatively small values is that behavior becomes highly stochastic with higher values. In our simulations, rewards were binary, with values 0 or 1. As a result, low values of β lead to much less exploitative behaviour than in situations where rewards can take larger values (as in Gaussian bandits with higher mean reward, e.g., Toyokawa & Gaissmaier, 2022).

In response to the reviewer's feedback, we have now conducted additional SFL model simulations with a wider range of parameter values. In Fig. R2-1 below, we plot predictions for Experiment 1 across a range of α and β values. This analysis shows that the predicted difference between the conditions at the outset of the Test phase, which was the crucial test of the SFL model (Fig. R2-1; red shaded area), gradually disappears as the values of α and β increase.

Fig. R2-1. A priori SFL model predictions for Experiment 1, across a range of α and β parameter values. The weight prior parameter was kept constant at 0.1. Blue lines indicate the Congruent condition and red lines the Incongruent condition. See Table S1 for estimated empirical parameter values from all experiments.

The main reason for the differences between conditions disappearing is that when Q-values (weight * feature value) are only based on the social feature (as in the outset of the Test phase), the difference between the options is always small (and smaller with a higher α). Specifically, the feature value is mean-centered, so for demonstrator choice proportions of, for example, 0.8:0.2, the feature value is 0.3:-0.3. This means that for a social feature weight of 0.1 (a typical value), the actual Q-values entering the Softmax choice function are 0.03 vs -0.03. The same logic extends to the role of the weight prior parameter at the very beginning of the task, which will have little impact for higher β values. In Fig. R2-2 below we illustrate the predicted choice probability difference for different values of β , given the example Q-values ($Q = -0.03$ and 0.03).

Fig. R2-2. The influence of the Beta parameter on the Softmax difference (i.e., the probability of selecting action A over action B) for a constant Q-value difference.

As a consequence, relatively small values for α and β are required to predict a behavioral difference between the Congruent and Incongruent condition at the outset of the test phase. However, participants' estimates do fall, on average, in this range (see also Table R2-1 below). For example, for Experiment 1, the median estimated α was 0.15, and the median estimated β was 0.146, i.e., highly similar to the values we used for simulation in the main text.

We now include Fig. R2-1 as Fig. S2 in the SI of the revised manuscript and reference it on p. 11 in the main text.

In the post-hoc agent-based model, the parameters were fixed at alpha = 0.2 and beta = 0.1, which could also bias the results. The authors should report results for a wider range of parameter values to cover the behavior of both myopic learners and explorative decision-makers.

Response: We have conducted additional simulations in which we vary α and β . To keep this manageable in scope, we limited this to the temporal change scenario, and to others' choice (Fig. R2-3) and payoff features (Fig. R2-4). Similar to the simulations discussed in the preceding response (cf. Fig. R2-1), we find that the simulated average feature weights tend to be smaller for higher values of α and β . This can be explained by the fact that when others' actions become more stochastic, they stop being reliable predictors of reward. As a consequence, social feature weights will be smaller. We now include figures R2-3 and R2-4 as Figs. S13 and S14 in the SI, and refer to them from the main text (p. 39): "see Fig. S13-S14 for simulations with alternative values for the learning parameters".

Fig. R2-3. The median simulated social feature weight W_{Others} , as a function of different values of the SFL parameters alpha (α) and beta (β). Each value represents the median of 800 simulation runs.

Fig. R2-4. The median simulated social feature weight W_{Payoff} , as a function of different values of the SFL parameters alpha (α) and beta (β). Each value represents the median of 800 simulation runs.

Furthermore, I am curious about the experimentally calibrated parameter values. Reporting the parameter fit results would enhance the robustness of their findings.

Response: We thank the reviewer for this comment and we believe many readers will appreciate adding this information. We now report the average estimated parameters for the SFL model (Table R2-1 below; Table S1 in the revised manuscript, referenced from p. 19 in the main text). In general, the average fitted parameters were similar to the ones used for simulations presented in the main text.

Experiment	Parameter	Min	25%	Median	75%	Max
1	Alpha	0	0.043	0.15	0.363	1
1	Beta	0.01	0.016	0.146	0.285	1
1	Prior	-2.016	-0.007	0.117	0.614	12.387
2	Alpha	0	0.048	0.158	0.37	1
2	Beta	0.01	0.019	0.127	0.278	1

2	Prior	-2.638	0.066	0.322	1.183	15.749
3	Alpha	0	0.003	0.109	0.472	1
3	Beta	0.01	0.07	0.306	0.617	1
3	Prior_Choice	-18.702	0.072	0.667	1.503	33.856
3	Prior_Payoff	-5.804	0.221	1.188	2.171	35.773
4	Alpha	0	0.081	0.208	0.374	1
4	Beta	0.01	0.102	0.17	0.258	1
4	Prior	-2.056	-0.008	0.237	0.667	20.137
5	Alpha	0	0.027	0.222	0.475	1
5	Beta	0.01	0.01	0.109	0.264	1
5	Prior	-7.603	0.003	0.2	1.127	95.965
6	Alpha	0	0.051	0.229	0.482	1
6	Beta	0.01	0.056	0.154	0.273	1
6	Prior	-24.65	-0.046	0.136	0.998	62.843

Table R2-1. Estimated SFL parameters in Experiments 1-6.

(2) Dual Alpha Model?

Related to the point above, I wonder why the authors did not test a model with two alphas (one for non-social feature updating and one for social feature updating). Testing such a dual-alpha model with human data, and ideally demonstrating that the single-alpha model is the winning model, would provide stronger evidence for the main finding that the same learning mechanism (with common alpha values) underlies both social and non-social feature learning. Currently, it is unclear if a single alpha suffices or if this result is due to the lack of consideration of multi-alpha models.

Response: We thank the reviewer for this thoughtful suggestion. To address this point, we estimated a dual-alpha model for each experiment (excluding the new two-feature Experiment 3 for simplicity, as it would require introducing several alphas for comparability). The dual-alpha model includes separate learning rates for social and non-social feature updating, as proposed by the reviewer.

Our results indicate that the single-alpha model fits better across all experiments (Fig. R2-5 below). Overall, these findings suggest that a single learning rate is sufficient to account for participants' behavior in most cases. Importantly, there is no compelling evidence from our data to support the necessity of a dual-alpha model.

In the revised manuscript, we mention that a single learning rate provides a parsimonious and sufficient account of the data (p. 33): “We also considered a version of the SFL model with separate α parameters for the non-social and the social feature, but this did not improve model fit.”

Fig. R2-5. Comparison of 1 and 2 alpha in the SFL model. The figure depicts the protected exceedance probability (bars) and posterior frequencies (dots) for the SFL model (with prior) with 1 or 2 alpha parameters.

(3) Multi-Armed Tasks?

Studying conformity/anti-conformity bias using a two-armed bandit could mislead conclusions because avoiding the majority always means following the minority. Previous literature has provided reasons why focusing solely on two-option tasks is inadequate (Muthukrishna et al., 2016). The authors should explore tasks with a greater number of options (such as the four-armed or eight-armed bandits used in the dangerous environment simulation).

Muthukrishna, M., Morgan, T. J., & Henrich, J. (2016). The when and who of social learning and conformist transmission. Evolution and Human Behavior, 37(1), 10-20.

Response: We thank the reviewer for highlighting this important issue, and for suggesting an extension of our basic paradigm to incorporate more choice options. In response, we conducted an additional experiment (n=353; pre-registration: https://aspredicted.org/PKH_DL1). This additional experiment featured two additional choice options, with intermediate feature values. Participants chose between four different options during the Learning phase. In the Congruent condition, the four options were chosen by 65%, 15%, 15%, and 5% of others, such that the majority choice was reward-predictive. This pattern was reversed in the Incongruent condition, such that the options were chosen by 5%, 15%, 15%, and 65% of others and the minority choice was reward-predictive. The Test phase introduced four novel choice options with 25% reward probability each, and which again were chosen by 65%, 15%, 15%, and 5% of others.

As before, we first generated predictions under the SFL model (see also Figure S5j in the revised manuscript, where predictions are generated based on the parameters estimated for Experiment 1). Mirroring the previous experiments, participants in the Congruent condition are predicted to follow the majority whereas participants in the Incongruent condition are predicted to follow the minority at the outset of the Test phase (Fig. R2-6a). The experimental results closely match these predictions: Participants in the Congruent condition chose the majority option more (73%) than participants in the Incongruent condition (14%, logistic regression of difference between conditions: $\beta = 2.86$, $SE = 0.33$, $z = 8.6$, $p < .001$, Fig. Fig. R2-6b). Similarly, participants in the Incongruent condition chose the minority option (50%) more than participants in the Congruent condition (10%, logistic regression of difference between conditions: $\beta = 2.26$, $SE = 0.35$, $z = 6.53$, $p < .001$).

a. Additional Experiment 1: Model Predictions

b. Additional Experiment 1: Behavioural Results

Figure R2-6. Additional Experiment 1. **a**, Majority choice in the new four-option experiment for the Congruent (blue lines) and Incongruent (red lines) condition, as predicted by the Social Feature Learning model. Faint lines represent single simulation runs and bold lines represent averages. The model predicts that participants in the Congruent condition learn to follow the majority and the Incongruent condition the minority, and this learning generalises to novel stimuli in the test phase (denoted by the vertical black line at trial 91). Each option had the same probability of being rewarded in the test phase, which explains the gradual convergence of the lines towards random choice (probability 0.25; horizontal line). **b**, Choice behaviour in the four-option experiment ($n = 353$). Participant data closely matches the predicted behaviour in the test phase.

These results closely matched the Social Feature Learning model predictions, and confirm that reward learning shapes social learning in settings with more than two choice options. Overall, this additional experiment demonstrates that our findings can be extended to more realistic scenarios with a larger number of behavioural options, addressing the reviewer's key concerns about the generalisability of our conclusions. We are grateful for the reviewer's suggestion for this study and believe that it makes our conclusions much stronger.

In addition to these new empirical results, we also conducted the suggested new simulations of the agent-based model with 4, 8, and 16 options, in addition to the simulations with 2 options we reported in the initial submission. We limited this analysis to simulations with

others' actions as the social feature, given that the prior literature suggests that the number of choice options primarily impacts majority-biased / conformist social learning (Nakahashi et al., 2012). We find that the average social feature weight (the weight of others' action, W_{Others}) is similar across these simulations (Fig. R2-7). This demonstrates that our simulation-based finding that social learning strategies emerge from the SFL model also holds for larger environments with more options. If anything, the average feature weight was somewhat higher when there were more options. The reason is that since the reward probability of each option was drawn independently at random in these simulations, environments with more options are more likely to have some options with a very high reward probability. In turn, this produces a higher social feature weight, since it means that others' actions can be good predictors of reward.

We have added Fig. R2-7 to the SI as Fig. S10 and reference from the caption of the main ABM figure (p. 24):

“See Fig. S10 for simulations of choice problems with more than two options, confirming that our results generalize to more complex settings.”

Fig. R2-7 Multiple choice options in the agent-based model. The figure displays the median value of W_{Others} across 800 simulation runs for each combination. The number of time steps was increased proportional to the number of arms (2 arms = 100 steps, 4 arms = 200 steps, 8 arms = 400 steps. For 16 arms, we increased to 1600 time steps, since preliminary simulations based on 800 steps showed that the average feature weight was still growing at 800 time steps).

Minor points:

(4) What was the non-social features?

They used a cover-story of "rabbit hunt" where the non-social feature was the size of the rabbit shown (s_{rabbit} in Eq. 1). However, according to the supplementary Fig. S1, the task looked just like a 2-armed bandit task. How exactly did they manipulate the non-social feature? The authors should explain it in the main text (maybe soon after the Eq. 1 where they showed an example of $s_{\text{others}} = 0.8$).

Response: The reviewer is correct that the basic task corresponds to a 2-armed bandit with social information. The non-social features were "one-hot encoded" (i.e., they took a value of 0 or 1). For example, the features of the left action were encoded as [1 0 0.8] and for the right action [0 1 0.2]. In this example, 80% of demonstrators would have selected the left option, and 20% the right option.

We have clarified the design of the experimental task, by updating Fig. 2 in the main text (included as Fig. R2-8 below).

Fig. R2-8 | Social learning is shaped by individual reward experience. **a**, Participants completed a probabilistic decision-making task. Choice options had non-social (square colour) and social features (Exp. 1: Others' choices (as illustrated in **a**); Exp. 2: Others' payoffs). During the Learning phase, participants could learn which of two options was most rewarding. Social features and rewards were aligned in the Congruent condition and misaligned in the Incongruent condition. The subsequent Test phase was used to assess whether social learning was shaped by rewards; participants were exposed to two new options (different coloured squares). **b**, SFL model predictions (identical for the two social features, see Methods) of how reward experience in the Learning phase shapes social learning in the Test phase (faint lines: individual simulation runs; solid lines: averages; see Methods for details and Fig. S2 for simulations across a range of parameters). The model includes a weight prior parameter, which allows for a majority preference on the first trial of the Learning phase. **c-d**, Empirical results closely align with the SFL model; with lines showing averages with 95% CIs. **c**, In Experiment 1, participants learned to follow the majority (minority) in the Congruent (Incongruent) condition in the Learning phase. **d**, In Experiment 2, participants learned to follow (avoid) the option with high payoffs for others in the Congruent (Incongruent) condition

in the Learning phase. For both Experiments, social learning spilled over into the Test phase. Grey boxes highlight predictions at the outset of the Test phase; *** indicates $p < .001$.

(5) Decision-biasing model

Their "fixed heuristic" model is also called "decision-biasing" in the literature. Perhaps it would help readers connect the current work with those previous works if the authors mention it when they introduce the term. (and the ref. 23 is also using the similar decision-biasing model though the authors skip citing it there).

Response: We have followed the reviewer's suggestion, by adding this name when introducing the fixed heuristics model (p. 19).

(6) Figure 6: why do the positions of initial points differ between panels?

The distribution of points at $x = 0$ differs between panels (except for the panel a and d, they are quite close). My understanding is that the focal feature was switched off when $x = 0$, which should make the balance between the four weights always the same among those panels. Could the authors clarify why the initial points differ?

Response: We thank the reviewer for this chance to clarify Figure 6 (now Figure 7). At $x = 0$ the environmental variable (i.e., the probability that reward probabilities change, the probability of migration, etc) is "switched off". The weight of a given feature should be about the same across the scenarios when $x=0$, as this means there is no difference between the scenarios.

We noticed an averaging error for the "age" feature in the code for the figure in the initial submission, which wrongly made the value of the age feature differ between the temporal/spatial/competition plots. We have fixed this for the revised figure. Any remaining difference between the $x=0$ value for a given feature between the panels is due to simulation stochasticity. Fig. S9 of the revision shows outcome distributions for each set of simulations to clarify this point.

The scenario where feature weights differ from the others at $x=0$ is "Danger", where weights consistently are higher. The reason for this is the same as why more actions result in, on average, higher feature weights (as discussed above, Fig. R2-7): since the reward probability of each option was drawn independently at random in these simulations, environments with more options are more likely to have at least one option with a high reward probability. In turn, this produces a higher social feature weight, since it means that others' actions can be good predictors of reward.

(7) Why did w_{others} not become negative?

In the competition manipulation (fig. 6d), w_{others} seems to converge to zero rather than a negative value. If copying "minority" helps avoid scramble competition, why does w_{other} not become negative?

Response: We thank the reviewer for this insightful remark. Two key factors contribute to this phenomenon:

First, increased competition results in smaller average absolute rewards for individuals, which in turn leads to smaller feature weights. In the extreme case where all rewards (and weights, as assumed for naive individuals in our simulations, see Methods) are zero, no learning can occur. However, it is important to note that our simulations do not include this extreme scenario.

Second, competition drives individuals to distribute themselves across the available choice options, consistent with the principles of the ideal free distribution (Fig. R2-9). As a result, the social feature values for each option become more similar, which slows down or stops the learning process. For instance, if individuals are exactly evenly distributed between two options, the feature values for those options will become identical (e.g., 0.5, which becomes 0 after mean-centering). Consequently, the social feature weight will converge to 0. This implies that when individuals disperse based on competition, there is less opportunity to use social features to infer or learn about rewards. We now clarify this in the revised manuscript (p. 26): “When individuals distribute themselves due to competition (Street et al., 2018), the social features of alternatives will tend to converge. For example, if an equal proportion of individuals selects each option, these options will have identical value of the feature related to others’ actions.”

Fig. R2-9 Depiction of the choice distribution of individuals across two options with different reward probability as a function of competition (Degree of competition: 0 vs 0.5).

(8) *Minor typo? Fig. 7 legend: "while the dotted horizontal line ..." should be "the dashed diagonal line" instead?*

Response: We thank the reviewer for pointing out this typo, which has been corrected.

Reviewer #3:

Remarks to the Author:

Review for social feature learning by

This study (by Schultner et al) proposes a reward learning framework to explain the emergence of social learning strategies. Through 4 experiments and multi-agent simulations they show that people adjust their learning based on rewards associated with social features like behavior and success and number of people (“majority”). There is much to like in this paper, starting from the fundamental question, and passing through the exaction and the open-science practices adopted by the authors.

Response: We thank the reviewer for their positive evaluation and constructive comments.

I do have, however, some questions, suggestions and remarks to share with the authors and the editors.

1/ First, I believe that a key experimental test of their theory would consist of a task where multiple features (i.e., number of subjects and payoff are simultaneously displayed to the participants, ideally in two versions (either payoff is predictive or number of subjects is predictive) to show that they model work in the more ecological situations where multiple features are presented. I believe this is partially addressed by the multi-agent simulations (that show that the models can work, in theory) but not empirically (in all experiments there is only one social feature). I am also aware that the authors already produced a lot for work, so I will defer to the editors’ opinion concerning how important this additional experiment could be.

Response: To test the Social Feature Learning in these ecologically relevant scenarios, we conducted a second additional experiment (n= 362 participants, pre-registration: https://aspredicted.org/DCD_YB3). In this experiment, participants were presented with choice options with two social features: others’ choices and others’ payoffs. Critically, only one feature correlated with reward. In the Learning phase of the Choice congruent condition, others’ choices were reward-predictive, while others’ payoffs were not. Conversely, in the Payoff congruent condition, others’ payoffs were reward-predictive, and others’ choices were not.

After the Learning phase, participants completed a Test phase where the two social features were pitted against each other: One option was chosen by the majority but yielded low payoffs to others, whereas the other option was chosen by the minority but yielded high payoffs to others. The SFL model posits that only reward-predictive features (and not features uncorrelated with rewards) should guide social learning.

As predicted by the SFL model (Fig. R3-1a), participants preferred the option aligned with the reward-predictive social feature in the first trial of the Test phase (Fig. R3-1b; logistic regression: $\beta = 1.53$, SE = 0.25, $z = 6.11$, $p < .001$). These results provide additional direct support for our model, using the experimental test suggested by the reviewer. In the revised manuscript, we have included this additional experiment as Experiment 3 and added Fig. R3-1 as panels to Fig. 3.

a. Additional Experiment 2: Model Predictions

b. Additional Experiment 2: Behavioural Results

Fig. R3-1. Additional Experiment 2. **a**, The probabilities of making the same choice as the majority (left) or the choice associated with others' higher payoff (right) in the new experiment with two social features (others' choices and others' payoffs), as predicted by the Social Feature Learning model (see also Fig. S2 d&g for predictions based on the parameters estimated in Experiment 1). On each trial of this experiment, participants faced two social features simultaneously (others' choices and others' payoffs), but only one feature correlated with reward. In one condition, the majority choice was reward-predictive, whereas the payoff feature was not. In the other condition, the payoff feature was reward-predictive, whereas others' choices were not. Each faint line represents a single simulation run and the bold lines represent the average behaviour. The vertical black line at trial 51 indicates the start of the test phase. Each option had the same probability of being rewarded in the test phase, which explains the gradual convergence of the lines. The horizontal line at 0.5 denotes random choice **b**., Choice behaviour in the second additional experiment ($n = 362$).

2/ The authors mainly focus on the simulations of their model. However, as readers, we are left without information concerning whether or not the other models were able to reproduce the same behavioural signatures (model falsification: see Palminteri et al. 2027) (I suspect that, to some extent, yes, especially the heuristics models - but this would not be the case in the 'two social features experiment' suggested above).

Response: We thank the reviewer for raising the issue of model falsification, and agree that this is crucial for the validation of our model. In response, we simulated Experiments 2-6

based on the median estimated parameters from Experiment 1. This allowed us to assess model generalization and falsification at the same time.

The results from this analysis are shown below (and included as Figure S7 in the revised manuscript). Figure R3-2 below demonstrates strong out-of-sample generalization for the SFL model, as the predicted results are qualitatively similar to the empirical results across all experiments (see also Fig. R3-6 & R3-7 below for quantitative out-of-sample predictions). In other words, the SFL model provides a generalizable account of social learning in these 6 experiments.

Our analysis also demonstrates that only the SFL model can reproduce the key patterns observed in the experimental data. For example, only the SFL model reproduces the difference between the Congruent and Incongruent conditions at the outset of the test phase in Experiment 1 & 2 (Fig. R3-2a). The Fixed Heuristics model does not include a mechanism that would allow the Congruent and Incongruent groups to differ in the Test phase (since social influence is fixed within individuals). While the Value shaping model could in theory create such a difference, this does not happen in practice.

Similarly, only the SFL model accurately captures the Test phase results of the new two-feature experiment (Exp. 3, Fig. R3-2d-i), and the results of the feature competition experiment (Exp. 6, Fig. R3-2m-o).

In other words, in the terminology of Palminteri (Palminteri et al., 2017), these results falsify the Fixed Heuristics and Value Shaping models and provide clear support for the SFL model.

Fig. R3-2. Out-of-experiment model simulations for all experiments. For all experiments, models were simulated based on the median estimated parameters from Experiment 1. (a-c) Experiment 2 (c.f. Figure 2, main text). Note that the models make the same predictions for Experiments 1 and 2 (and the Learning phase of Experiment 5). (d-i) Model simulation of Experiment 3 (c.f., Figure 3, main text), displayed separately for the Choice congruent and the Payoff congruent condition. (j-l) Model simulation of Experiment 4 (c.f., Figure 3, main text). (m-o) Model simulation of the Transfer phase of Experiment 6. Dots show the predicted probability of choosing the single condition color for the high and low reward value pairs (cf., Figure 4, main text). The SFL and Value Shaping models both include prior parameters. For the payoff feature in Experiment 4, the estimated prior parameters from Experiment 2 were used. The shaded red areas in panels c, f, i, and l denote the Test phase, where two new choice options were introduced.

We want to clarify two points concerning the model comparison in the revised manuscript. First, we have decided to remove the permutations of the Fixed Heuristics models (i.e., hybrid models with learning) from both the model comparison and model simulation analyses. By removing these model variations, our manuscript now focuses on models with a strong theoretical rationale and avoids diluting the distinctiveness of the model families (i.e., Fixed Heuristics/decision-biasing, Value Shaping, and Social Feature Learning).

Second, we now select the best model within each family (SFL and Value shaping. Each of these families was represented by two models, with and without a prior respectively. The Fixed Heuristics family was represented by only one model) in each experiment as a basis for model comparison. Our reasoning was that this prevents models within families from competing (since their fit will typically be similar) and further focuses the presentation.

We describe this in the revised methods section (p. 39): “Because there were two versions of both the SFL and Value Shaping models (with and without a prior parameter), we first assessed, for each experiment, which version provided the best fit, and used this version for the overall model comparison.”

Additionally, we noted that for Experiment 1 (but no other experiments), the range of the SFL prior parameter was wrongly restricted in the model fitting script used for the original submission. We find that, after removing this restriction, the SFL+prior fits slightly better than the basic SFL model for Exp.1. In total, the SFL+prior fits better than the basic SFL model in all experiments but Exp. 5 (previously referred to as Exp. 4).

Similarly, what is the model recovery/confusion matrix?

Response: We thank the reviewer for raising this issue. In response, we have added a range of measures addressing model identifiability, including confusion matrices, inversion matrices, and out-of-sample predictions (Palminteri et al., 2017; Wilson & Collins, 2019) Our overall goal here was to assess whether our models of interest are distinguishable in the behavioral patterns they predict. Below, we outline the steps taken to address the concern in detail.

First, we conducted a model recovery analysis to test whether the model used to simulate the data would also provide the best fit to that data (Wilson & Collins, 2019). We based our analysis on Experiment 1, which serves as the basic test of the SFL model. To streamline the analysis, we focused on the best-fitting models from the three model families of interest: SFL (both with and without the Prior parameter, as these models performed similarly), Fixed Heuristics, and Value Shaping (without Prior).

We first simulated data from these three models, by randomly sampling parameter values (with replacement) from the set of estimated parameters. We generated 2000 synthetic participants from each model. The same fitting procedure used for the original data—individual-level maximum likelihood estimation followed by Bayesian random-effects model comparison—was applied to these simulated datasets.

First, we constructed a standard confusion matrix (Fig. R3-3a). This matrix shows the “probability that data generated by one model is best fit by another”. The results indicate

perfect recovery for the SFL model (here first without Prior), though the SFL model also fits data generated by the Fixed Heuristics and Value Shaping models reasonably well. One interpretation of the fact that the SFL model fits all simulations reasonably well is that its parsimony and flexibility allow it to assume the behavior of a range of models.

Fig. R3-3. Confusion and inversion matrices. (a) Confusion matrix. The numbers in the cells indicate the probability that data generated by one model is best fit by another. (b) Inversion matrix. The numbers in the cells provide the probability that the best fitting model is the true model. The analysis was based on simulations of Experiment 1. The SFL model did not have a prior parameter.

Although the confusion matrix indicates perfect recovery with regards to the SFL model, the status of the Fixed Heuristics and Value Shaping models remains ambiguous. In response, we conducted a second step: We created the “inversion matrix” (Wilson & Collins, 2019), which addresses the more intuitive question: “Given that model B fits the data best, which model is most likely to have generated the data?” This metric is particularly relevant when the true generating model is unknown, as in empirical applications. The inversion matrix is generated by normalizing the columns of the confusion matrix (Wilson & Collins, 2019).

Inspection of the inversion matrix (Fig. R3-3b) reveals that, importantly, the values are maximised on the diagonal, suggesting that each model is most likely to have generated itself (conditional on fitting the data best). However, the SFL model only shows a ~0.5 inversion probability, suggesting some mimicry among the models. To assess the reliability of this probability, we computed bootstrap confidence intervals of the inversion matrix (calculated based on 1000 samples of 250 participants each). This analysis consistently yields inversion probabilities for the SFL model around ~0.5, and the 95% CI [0.46, 0.51] did not overlap with chance (~0.34), indicating that the SFL inversion probability is reliable.

For the SFL model with Prior, the confusion and inversion probabilities (see Fig. R3-4) are slightly lower, likely because the additional Prior parameter allows this variant to fit more extreme patterns that the other models may generate. Nonetheless, bootstrap confidence intervals of the inversion matrix demonstrate the inversion probability of the SFL+prior model

is reliably higher than chance (bootstrap mean = 0.41, 95% CI [0.39, 0.44], vs chance P = 0.34), again indicating that this difference is reliable

Fig. R3-4 Confusion and inversion matrices. (a) Confusion matrix. The numbers in the cells indicate the probability that data generated by one model is best fit by another. (b) Inversion matrix. The numbers in the cells provide the probability that the best fitting model is the true model. The analysis was based on simulations of Experiment 1. The SFL model included a prior parameter.

We have included Fig. R3-3 and Fig. R3-4 as Fig. S15 and Fig. S16 in the revised manuscript, and referenced in the main text (p. 39): “Model identifiability analyses were conducted by simulating data from each model based on the parameters of Experiment 1, and then fitting each model to each simulated dataset. If a model best fits the dataset it generated, it can be said to be identifiable. Section S3 explains this approach and displays its results.”. We have added a new section (S3) to the SI which details the model recovery, and the generalization approach we describe below.

To evaluate the validity of the SFL model beyond model recovery analyses, we tested its generalisability by generating out-of-sample predictions. Generalization outside the original experimental context is a robust, gold-standard measure of model validity, since “The best models don’t just explain data in one experiment, they *predict* data in completely new situations.” (Wilson & Collins, 2019). Our approach follows the generalization criterion method (Busemeyer & Wang, 2000): Based on the mean and median parameters estimated from Experiment 1, we calculated, per participant and model, the out-of-sample log-likelihood for Experiments 2-5. This approach does not entail any additional model fitting for Experiments 2-5. A benefit of this approach is that overfitted models will perform worse, as, by definition, such models will only fit the training sample (Experiment 1) well. Hence, if the SFL model overfits in-sample data and therefore mimics other models, this will result in poor out-of-sample predictions. On the contrary, analyses show that the SFL demonstrates successful out-of-sample performance (Fig. R3-5b), highlighting the SFL model’s ability to generalize across different situations. This indicates that the SFL model did not overfit the in-sample data.

We have included the out-of-sample tests in the main text of the revised manuscript (p. 19):

“In addition to quantitative model comparison, we also tested the generalisability of the SFL model by generating out-of-sample predictions. This involved using parameter estimates derived from Experiment 1 (see Table S1) to predict the data from Experiments 2–6, (see Methods). Employing such out-of-sample prediction provides a stringent test of a model's generalisability and robustness (Busemeyer & Wang, 2000; Palminteri et al., 2017). The SFL model provided the best overall out-of-sample predictions (Fig. 5b [reproduced as Fig. R3-5 below]), see also Fig. S6 for predictions for individual experiments and Fig. S7 for out-of-sample predictions based on the mean, rather than median, parameters), highlighting the SFL model's ability to generalize across different situations.”

Fig. R3-5 | In-sample model comparison and out-of-sample prediction. (a) The figure shows the estimated in-sample model probability (protected exceedance probability) of each model averaged across Experiments 1-6, representing the probability that this model is the best account of most participants' behavior. Dots indicate the posterior model frequencies, the estimated prevalence of each model. See Fig. S5 for model comparison separately for each experiment. (b) Out-of-sample model probability and posterior frequencies averaged across Experiments 2-6, based on the median estimated parameter values from Experiment 1. See Methods for details.

Comparing the out-of-sample predictions of the SFL and Value Shaping models based on median parameter estimates from Experiment 1 indicates comparable performance for Experiments 4 & 6 (Fig. R3-6), a comparison based on the mean estimated parameters clearly favored the SFL model (Fig. R3-7). Together, these results provide strong evidence that the SFL model provides the best generalization performance across experiments.

Fig. R3-6 | Out-of-sample predictions. Model probability (bars) and posterior frequencies (dots) for Experiments 2-6, based on the median estimated parameter values from Experiment 1.

Fig. R3-7 | Out-of-sample predictions. Model probability (bars) and posterior frequencies (dots) for Experiments 2-6, based on the mean estimated parameter values from Experiment 1. The top left panel depicts the average.

Second, and as discussed in our previous answer, we simulated all experiments based on the median parameters from Experiment 1, and found that only the SFL model could reliably

recreate the key empirical results (Fig. R3-2). As described above, only the SFL model reproduces the difference between the Congruent and Incongruent groups at the outset of the test phase in Experiment 1 - 2 & 4 (Figure R3-2a). Similarly, only the SFL model accurately captures the Test phase results of the new two-feature experiment (Exp. 3, Fig. R3-2 d-i), and the results of the feature competition experiment (Exp. 6, Fig. R3-2 m-o).

Together, the combined recovery, generalization, and falsification results provide clear evidence for the validity of the SFL model.

3/ I think that the authors are missing a very relevant literature here, specifically concerning Rieskamp's work on strategy selection:

<https://psycnet.apa.org/buy/2006-06642-005>

some of these studies propose precise models about how associative learning mechanisms may explain how heuristics are selected. Of course, there are many differences, but still I believe this literature is relevant.

Response: We thank the reviewer for highlighting this. In our revision, we have cited the seminal paper by Rieskamp and Otto (2006) (p. 22). We feel that this addition nicely connects our work to relevant models on strategy selection in adjacent research fields, and highlights how our feature-based model differs from strategy selection approaches.

4/ I wonder how much the authors really commit to the idea that I will summarize in this sentence: "There are no hard-coded social learning heuristics, they are all learned by trial-and-error from rewards". This is probably an extreme, unnuanced version but the authors seem nonetheless seem to lean toward this view. Now, the problem here is that I do not believe that this is fully true. We all have the intuition that social learning is automatically deployed before any reward is experienced (i.e., in childhood) and in many situations in which social learning, imitation, is used rewards are delayed. Can the authors comment on this and discuss these points?

Response: We agree that the manuscript would benefit from clearly pointing out how we see the relative importance of "fixed / hard-coded" versus "flexible / learned" factors in social learning. In the discussion (p. 30), we now clarify our position:

"In our experiments, participants quickly adapted their social learning strategies to changes in the reward associations of social features. Outside of the lab, this flexibility is likely to produce significant variability within and between individuals in what social learning strategies they use, and their reliance on social versus individual forms of learning. For example, individuals may develop and adopt a wide range of social learning strategies based on their unique experiences and the specific rewards they encounter. However, it should be emphasised that social feature learning does not exclude in-born biases in the initial weights of social features, and how the features are processed. These biases might facilitate effective learning, and are likely to be subject to natural selection (Croston et al 2015). The SFL model provides an account of how reward experiences may shape and refine these biases throughout development, attuning individuals' social learning to their environment."

We also agree with the reviewer that delayed rewards are common. The SFL model does not require that rewards are provided instantaneously after actions. For the sake of simplicity, our simulations and experiments considered situations in which rewards directly follow actions (although the actual time interval between the two is not specified in the simulations). However, the SFL model can be extended to sequential settings, by using a temporal-difference updating rule. This would allow the model to be influenced by social features, even though rewards are delayed by multiple actions or by the passing of time. In other words, we do not believe there is any tension between the fact that rewards can be delayed and our computational model.

We have added a note about this possible model extension to the discussion (p. 32):

“Moreover, although the experiments conducted here test scenarios with immediate feedback, the SFL model can naturally be extended to scenarios with delayed rewards: Employing a temporal-difference updating rule (Sutton & Barto, 1998) would allow social feature learning even when rewards are delayed by sequences of actions or by the passing of time.”

5/ Finally, I believe that the Figures could be improved. Figure 1 could be a bit clearer (what the grey vs. white mean? the legend mentions rectangles, but there is no rectangle, sensu strictu, in the figure). Maybe adding the equations? Figure 2 some of the axes are without legend (maybe because it is the same as on the right column). Plus ($p(\text{choice})$) is not very clear. More generally they could be made more informative.

Response: We agree with the reviewer that Figures 1 and 2 could be clearer. In Figure 2, we now refer to the rounded rectangles as ‘grey boxes’ and describe their purpose in the figure caption.

We have split up Figure 2 in the original submission into two figures (Figures 2 and 3 in the revision). We further cleaned up Figure 2 by (i) replacing the conceptual panel a) with a task depiction, (ii) reducing the figure by one row of panels, and (iii) adding a more descriptive figure caption. We have adopted the convention of dropping the y-axis label on the right panel if it is equivalent to that of the left panel, and replaced the axis labels with more descriptive ones (e.g., ‘P(Majority Choice)’). We believe our edits have substantially improved Figure 2, which we include below as Fig. R3-8.

Finally, we hope the reviewer is okay with us not showing equations in the figure (as was their suggestion), to avoid deterring non-mathematically-minded readers. We believe that overall, the reviewer’s suggestions have led to substantially improved Figures 1-3.

Fig. R3-8 | Social learning is shaped by individual reward experience. **a**, Participants completed a probabilistic decision-making task. Choice options had non-social (square colour) and social features (Exp. 1: Others' choices (as illustrated in **a**); Exp. 2: Others' payoffs). During the Learning phase, participants could learn which of two options was most rewarding. Social features and rewards were aligned in the Congruent condition and misaligned in the Incongruent condition. The subsequent Test phase was used to assess whether social learning was shaped by rewards; participants were exposed to two new options (different coloured squares). **b**, SFL model predictions (identical for the two social features, see Methods) of how reward experience in the Learning phase shapes social learning in the Test phase (faint lines: individual simulation runs; solid lines: averages; see Methods for details and Fig. S2 for simulations across a range of parameters). The model includes a weight prior parameter, which allows for a majority preference on the first trial of the Learning phase. **c-d**, Empirical results closely align with the SFL model; with lines showing averages with 95% CIs. **c**, In Experiment 1, participants learned to follow the majority (minority) in the Congruent (Incongruent) condition in the Learning phase. **d**, In Experiment 2, participants learned to follow (avoid) the option with high payoffs for others in the Congruent (Incongruent) condition

in the Learning phase. For both Experiments, social learning spilled over into the Test phase. Grey boxes highlight predictions at the outset of the Test phase; *** indicates $p < .001$.

6/ Was social information simulated, was this told to the participants or did the study involve actual social information?

Response: We agree with the reviewer that this issue is currently ambiguous. To allow for tight experimental control, the social information was simulated. Participants were led to believe that the social features represented the actual choices of past participants. After each experiment, participants were debriefed regarding the true nature of the social information they received. We have added an explanation of this to the Methods section (p.34 & 35):

“Although participants received the instruction that the demonstrators were past participants, demonstrators were in fact computer-controlled, and programmed to select each option with specific probabilities (see below). We sacrificed a deception-free design for the experimental control required to test our hypotheses.”

References

- Arora, S., & Doshi, P. (2019). *A Survey of Inverse Reinforcement Learning: Challenges, Methods and Progress*. <https://arxiv.org/pdf/1806.06877.pdf>
- Bussemeyer, J. R., & Wang, Y. M. (2000). Model comparisons and model selections based on generalization criterion methodology. *Journal of Mathematical Psychology*, *44*(1), 171–189.
- FeldmanHall, O., & Nassar, M. R. (2021). The computational challenge of social learning. *Trends in Cognitive Sciences*, *25*(12), 1045–1057.
- Gweon, H. (2021). Inferential social learning: Cognitive foundations of human social learning and teaching. *Trends in Cognitive Sciences*, *25*(10), 896–910.
- Heyes, C. (2016). Blackboxing: social learning strategies and cultural evolution. *Philosophical Transactions of the Royal Society of London. Series B, Biological Sciences*, *371*(1693), 20150369.
- Jara-Ettinger, J. (2019). Theory of mind as inverse reinforcement learning. *Current Opinion in Behavioral Sciences*, *29*, 105–110.
- Jiménez, Á. V., & Mesoudi, A. (2019). Prestige-biased social learning: current evidence and outstanding questions. *Palgrave Communications*, *5*(1), 1–12.
- Kendal, R. L., Boogert, N. J., Rendell, L., Laland, K. N., Webster, M., & Jones, P. L. (2018). Social Learning Strategies: Bridge-Building between Fields. *Trends in Cognitive Sciences*, *22*(7), 651–665.
- Leong, Y. C., & Zaki, J. (2018). Unrealistic optimism in advice taking: A computational account. *Journal of Experimental Psychology. General*, *147*(2), 170–189.
- Mesoudi, A., Chang, L., Dall, S. R. X., & Thornton, A. (2016). The Evolution of Individual and Cultural Variation in Social Learning. *Trends in Ecology & Evolution*, *31*(3), 215–225.
- Mondragón, E., Alonso, E., & Kokkola, N. (2017). Associative Learning Should Go Deep. *Trends in Cognitive Sciences*, *21*(11), 822–825.

- Nakahashi, W., Wakano, J. Y., & Henrich, J. (2012). Adaptive Social Learning Strategies in Temporally and Spatially Varying Environments: How Temporal vs. Spatial Variation, Number of Cultural Traits, and Costs of Learning Influence the Evolution of Conformist-Biased Transmission, Payoff-Biased Transmission, and Individual Learning. *Human Nature (Hawthorne, N.Y.)*, 23(4), 386–418.
- Niv, Y., Daniel, R., Geana, A., Gershman, S. J., Leong, Y. C., Radulescu, A., & Wilson, R. C. (2015). Reinforcement Learning in Multidimensional Environments Relies on Attention Mechanisms. *The Journal of Neuroscience: The Official Journal of the Society for Neuroscience*, 35(21), 8145–8157.
- Palminteri, S., Wyart, V., & Koechlin, E. (2017). The Importance of Falsification in Computational Cognitive Modeling. *Trends in Cognitive Sciences*, 21(6), 425–433.
- Radulescu, A., Shin, Y. S., & Niv, Y. (2021). Human Representation Learning. *Annual Review of Neuroscience*, 44(1), 253–273.
- Ray, E., & Heyes, C. (2011). Imitation in infancy: the wealth of the stimulus. *Developmental Science*, 14(1), 92–105.
- Siegel, J., Mathys, C., Rutledge, R., & Crockett, M. (2018). Beliefs about bad people are volatile. In *PsyArXiv*. <https://doi.org/10.31234/osf.io/2cqkz>
- Street, G. M., Erovenko, I. V., & Rowell, J. T. (2018). Dynamical facilitation of the ideal free distribution in nonideal populations. *Ecology and Evolution*, 8(5), 2471–2481.
- Sutton, R. S., & Barto, A. G. (1998). Reinforcement Learning: An Introduction. *IEEE Transactions on Neural Networks / a Publication of the IEEE Neural Networks Council*, 9(5), 1054–1054.
- Toyokawa, W., & Gaissmaier, W. (2022). Conformist social learning leads to self-organised prevention against adverse bias in risky decision making. *eLife*, 11, e75308.
- Vélez, N., & Gweon, H. (2021). Learning from other minds: An optimistic critique of reinforcement learning models of social learning. *Current Opinion in Behavioral Sciences*, 38, 110–115.
- Wilson, R., & Collins, A. (2019). Ten simple rules for the computational modeling of behavioral data. *eLife*. <https://doi.org/10.31234/OSF.IO/46MBN>
- Wise, T., Emery, K., & Radulescu, A. (2024). Naturalistic reinforcement learning - PubMed. *Trends in Cognitive Science*, Feb 28, 144–158.